# TEDM: TIME SERIES FORECASTING WITH ELUCIDATED DIFFUSION MODELS

**Edgardo Solano-Carrillo, Sreerag V Naveenachandran, Julia Niebling**
Institute of Data Science, German Aerospace Center (DLR)
`{edgardo.solanocarrillo, sreerag.naveenachandran, julia.niebling}@dlr.de`

## ABSTRACT

Score-based generative modeling through differential equations has driven breakthroughs in high-fidelity image synthesis, offering modular model design and efficient sampling. However, this success has not been widely translated to time-series forecasting yet. This gap stems from the sequential nature of time series, in contrast to the unordered structure of images. Here, we extend the theoretical formulation used for images to explicitly address sequential structures. We propose a diffusion-based forecasting framework (TEDM) that adapts score estimation to temporal settings and elucidates its design space. Such a design allows empirical computation of noise and signal scaling directly from data, avoiding external schedules. Notably, this reduces sampling complexity to linear in the forecast horizon. Without elaborate preprocessing, TEDM sets new state-of-the-art results on multiple forecasting benchmarks. These results illustrate the growing potential of diffusion models beyond vision. TEDM generates low-latency forecasts using a lightweight architecture, making it ideal for real-time deployment.

## 1 INTRODUCTION

Multivariate time-series forecasting drives critical decision-making across domains as varied as demand planning (Kamarthi et al. (2024)), financial risk assessment, weather prediction (Oskarsson et al. (2024), stock market analysis (Zou et al. (2023)), and energy load forecasting (Symeonidis & Nikolaidis (2025)). Unlike classical regression tasks, time-series data exhibit unique characteristics—trend, seasonality, and autocorrelation—that demand models capable of capturing temporal dependencies and quantifying predictive uncertainty, particularly in high-stakes settings such as meteorology and finance (Box et al. (2015)).

Recent advances in deep learning have dramatically improved forecasting accuracy by leveraging sequence models. Transformer-based architectures in particular—such as Informer (Zhou et al. (2021)) and Autoformer (Wu et al. (2021))—have consistently topped benchmark leaderboards. However, these approaches exhibit high computational complexity (quadratic time and memory requirements) (Kim et al. (2024); Kong et al. (2025)), and have poor long-term forecasting performance.

Diffusion models have emerged as a powerful generative paradigm across modalities, achieving state-of-the-art results in image, speech, and video synthesis (Xing et al. (2024); Ahsan et al. (2025)). Early attempts to adapt diffusion modeling to time series, like TimeGrad (Rasul et al. (2021)), showed promise in computational complexity, by processing sequences using recurrent networks instead of transformers. Nevertheless, they fall short in forecasting long horizons. Other approaches followed (Yang et al. (2024); Su et al. (2025)), demonstrating that longer horizons are possible at the expense of more preprocessing and model complexity, additional to the inherent sampling inefficiency of diffusion models (Ma et al. (2025)).

To strike a balance between long-term forecasting performance and computational complexity, a deeper understanding of the design space of diffusion models is needed. In the vision domain, this was elucidated by the EDM framework of (Karras et al. (2022)) and imported to time series forecasting in climate applications (Price et al. (2025)). This allows optimization of noise and scale schedules, as well as time-discretization strategies and solvers for the diffusion process. However, the sampling inefficiency remains, scaling as $O(SH)$ for $S$ diffusion steps and $H$ forecasting steps.

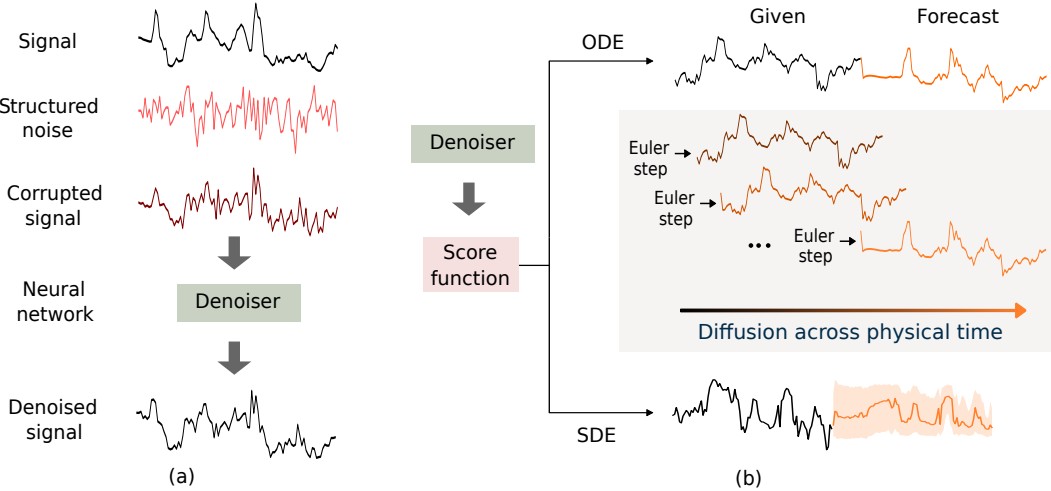

Figure 1: (a) Denoising signals corrupted with structured noise. (b) TEDM uses score-based generative modeling to forecast time series by numerical integration of an ordinary (ODE) or stochastic (SDE) differential equation. The main characteristic of TEDM is reducing the physical time and the diffusion time to the same axis, as we support by theory.

To address these limitations, we propose TEDM (**T**ime Series Forecasting with **E**lucidated **D**iffusion **M**odels), an autoregressive diffusion framework tailored for multivariate probabilistic forecasting. We extend the theoretical background of EDM for time series forecasting. Optimization of the design space thus leads to significant reduction in complexity and increase in accuracy by:

- Treating the diffusion and physical time axes as the same (see Fig. 1b). This reduces sampling complexity from $O(SH)$ to $O(H)$.

- Using, for the first time, noise and scale schedules estimated empirically from the data. This avoids inductive biases from guiding the diffusion with artificially imposed schedules.

This allows TEDM to achieve state-of-the-art results on several long-sequence forecasting benchmarks at a fraction of the cost of traditional methods. We find that the best case for space complexity is $O(T\,d)$ for a given sequence of $T$ timesteps and $d$ features—still giving promising results and then making our approach suitable for online settings.

## 2 RELATED WORK

**Sequence Models**. RNNs (Hewamalage et al. (2021)) and TCNs (Chen et al. (2020)) have been applied to capture nonlinear, temporal dependencies. Recently, Transformer-based variants—Informer (Zhou et al. (2021)), Autoformer (Wu et al. (2021)), DLinear, and iTransformer (Liu et al. (2024))—tackle long-range interactions via attention mechanisms and are dominating leaderboards. However, attention on sparse or irregularly-sampled timestamps can degrade, and preserving temporal ordering remains challenging (Wu et al. (2021)). Additionally, most Transformer-based forecasters produce only point estimates, limiting uncertainty quantification.

**Diffusion models**. Popularized by Score Matching with Langevin Dynamics (SMLD) (Song & Ermon (2020)) and Denoising Diffusion Probabilistic Models (DDPM) (Ho et al. (2020)), they are promising for probabilistic time series forecasing due to their generative nature. TimeGrad (Rasul et al. (2021)) pioneered autoregressive score-based forecasting by combining RNN encoders with per-step diffusion sampling, though it inherits RNNs' inefficiencies over long horizons. Non-autoregressive variants—such as TimeDiff (Shen & Kwok (2023)) and TSDiff (Kollovieh et al. (2023)) generate the prediction horizon in one step—bypassing error accumulation and enabling parallel forecasting. ARMD (Gao et al. (2025)) improves autoregression and sampling complexity by supervising a devolution network that effectively learns to "jump" the $S$ diffusion steps.

Despite substantial advances, most of the existing diffusion-based forecasters are merely adaptations of image-domain DDPMs that do not completely harness the multivariate and temporal structures of time-series data. This is due to an incomplete knowledge of the full design space of diffusion models that EDM helps to elucidate. In the following, we describe this background and introduce the changes needed for time series forecasting ultimately leading to TEDM.

## 3 BACKGROUND

### 3.1 EDM: UNIFIED DESIGN SPACE FOR DIFFUSION MODELS

Karras et al. (2022) present a unified framework for analyzing and improving diffusion-based generative models for image synthesis. Their core contribution is the disentanglement of architectural, training, and sampling components into a modular design space, enabling independent optimization of each element. This allowed them to find optimal choices for each component and push the state of the art for image synthesis.

**Deterministic sampling**. In their formulation, sampling is grounded on the probability flow ordinary differential equation (ODE)

$$\frac{d\boldsymbol{x}_t}{dt} = \frac{\dot{s}_t}{s_t}\,\boldsymbol{x}_t - s_t^2\,\dot{\sigma}_t\sigma_t\nabla_{\boldsymbol{x}}\log p_t(\boldsymbol{x}_t). \tag{1}$$

Here, $\boldsymbol{x}_t \in \mathbb{R}^d$ is the sample, $\sigma_t$ is a time-dependent noise schedule, $s_t$ is a time-dependent scale schedule, and $p_t(\boldsymbol{x}_t) = s_t^{-d}p(\boldsymbol{x}_t/s_t; \sigma_t)$ is the marginal distribution of the diffusion process. The latter is expressed in terms of a mollified version of the data distribution obtained by adding i.i.d Gaussian noise—of standard deviation $\sigma_t$—to the samples. The term $\nabla_{\boldsymbol{x}}\log p_t(\boldsymbol{x}_t)$ is called the *score function* (Song et al. (2021)).

Deterministic sampling is achieved by integrating the ODE backwards, from time $T$ where $\boldsymbol{x}_T$ is completely noisy ($\sigma_T$ maximum), to time 0 where $\boldsymbol{x}_0$ is the prediction ($\sigma_0 \sim 0$). A related stochastic differential equation (SDE) adds noise during sampling for improved robustness. The authors Karras et al. (2022) propose using a second-order Heun's method with a linear schedule $\sigma_t = t$ and constant scaling $s_t = 1$, which leads to smoother sampling trajectories.

**Denoising score matching**. Training is based on denoising score matching: given clean data $\boldsymbol{y} \sim p_{\text{data}}$ and Gaussian noise $\boldsymbol{\varepsilon} \sim \mathcal{N}(\boldsymbol{0}, \boldsymbol{I})$, the data is corrupted with *unstructured* noise $\boldsymbol{n} = \sigma_t\boldsymbol{\varepsilon}$. The training objective then minimizes the expected value:

$$\mathbb{E}_{\boldsymbol{y}\sim p_{\text{data}},\boldsymbol{\varepsilon}\sim\mathcal{N}(\boldsymbol{0},\boldsymbol{I})}\left[\|D(\boldsymbol{y}+\boldsymbol{n};\sigma_t) - \boldsymbol{y}\|^2\right]. \tag{2}$$

This loss encourages the denoiser $D(\boldsymbol{x};\sigma_t)$ to estimate the conditional expectation of the clean signal given the noisy input $\boldsymbol{y}+\boldsymbol{n}$ as well as the noise level $\sigma_t$. It is related to the score function (see Eq. (3) of Karras et al. (2022)) by:

$$\nabla_{\boldsymbol{x}}\log p(\boldsymbol{x};\sigma_t) = \frac{D(\boldsymbol{x};\sigma_t) - \boldsymbol{x}}{\sigma_t^2}. \tag{3}$$

**Preconditioning**. To improve stability and expressiveness, the authors propose a preconditioned architecture for the denoiser $D_\theta$:

$$D_\theta(\boldsymbol{x};\sigma) = c_{\text{skip}}(\sigma)\,\boldsymbol{x} + c_{\text{out}}(\sigma)\,F_\theta(c_{\text{in}}(\sigma)\,\boldsymbol{x}; c_{\text{noise}}(\sigma)).$$

Here, $F_\theta$ is the core neural network to be trained, and $c_{\text{skip}}, c_{\text{in}}, c_{\text{out}}, c_{\text{noise}}$ are scalar functions of $\sigma$ that control signal scaling and conditioning. These are derived analytically by requiring the network inputs and training targets to have unit variance ($c_{\text{in}}, c_{\text{out}}$), and amplifying errors in $F_\theta$ as little as possible ($c_{\text{skip}}$). Except for $c_{\text{noise}}$ (which is chosen empirically), these functions are expressed in terms of $\sigma_{\text{data}}^2 = \text{Var}(\boldsymbol{y})$, which is uniformly set in image datasets. The authors also propose a log-normal distribution for sampling noise levels $\sigma_t$ during training, along with loss weighting $\lambda(\sigma) = 1/c_{\text{out}}^2(\sigma)$. This modular reformulation facilitates targeted improvements and enhances compatibility with a range of generative architectures.

Importantly, this unified framework subsumes earlier methods like DDPM, DDIM, and SMLD (Song et al. (2021)). These differ mainly in their $\sigma_t$ and $s_t$ schedules, time discretizations, and

preconditioning schemes. For instance, DDPM uses $\sigma_t = t$, $s_t = 1$ (as EDM), and stochastic sampling. DDIM replaces the stochastic reverse process with Euler integration of the same ODE and time steps as DDPM. SMLD instead models the score function directly with a variance-exploding SDE. This modular view reveals that improvements in training or sampling can often be transferred across models without retraining the network.

### 3.2 EXTENDING EDM TO MULTIVARIATE SERIES

Adapting EDM to multivariate time-series introduces unique challenges not present in image domains. First, applying a shared noise schedule across all features assumes uniform scale and dynamics. In practice, features may differ in variance or predictive importance, making uniform noise injection suboptimal. Feature-specific noise scaling is needed. Second, time-series exhibit strong temporal dependencies. EDM's i.i.d. Gaussian noise assumption can disrupt autocorrelated patterns, especially when noise is added independently at each time step. Structured noise or causal conditioning mechanisms may be required to preserve temporal coherence. Finally, architectural design must respect the sequential nature of time. Preconditioning schemes should account for temporal scale and position. Temporal encodings, autoregressive models, or attention-based architectures may be better suited than standard convolutional backbones. We discuss next how these challenges are theoretically and experimentally addressed by TEDM.

## 4 TEDM METHODOLOGY

### 4.1 PROBLEM DEFINITION

Given a multivariate series $\mathbf{y}_{1:T} \in \mathbb{R}^{C \times T}$, with $C$ features and $T$ time steps, the problem is to forecast the next $H$ steps. The forecast is done through a mapping $f_\theta : \mathbf{y}_{1:T} \mapsto \widehat{\mathbf{y}}_{T+1:T+H}$, consisting of a learned estimator of the score function and its autoregressive use in an ODE (or SDE) solver. Unless otherwise necessary, we omit the subscript from the respective variables. The theoretical results in the following are derived in appendix A.

### 4.2 DATA-DRIVEN NOISE AND SCALE SCHEDULES

We extend the EDM formulation to multivariate noise schedules. With $\mathbf{\Sigma}_t := s_t^{-2} \mathrm{Cov}(\boldsymbol{x}_t)$, the forward ODE in (1) takes now the more general form

$$\frac{d\boldsymbol{x}_t}{dt} = \frac{\dot{s}_t}{s_t} \boldsymbol{x}_t - \tfrac{1}{2} s_t^2 \, \dot{\mathbf{\Sigma}}_t \nabla_{\boldsymbol{x}} \log p_t(\boldsymbol{x}_t). \tag{4}$$

We restrict to deterministic sampling in the following and leave probabilistic forecasts (based on our SDE (A.65)) for Appendix D. In our formulation, the score function becomes (Eq. (A.25))

$$\nabla_{\boldsymbol{x}} \log p_t(\boldsymbol{x}_t) = s_t^{-1} \mathbf{\Sigma}_t^{-1} \big[ D(\boldsymbol{x}_t/s_t; \mathbf{\Sigma}_t) - \boldsymbol{x}_t/s_t \big], \tag{5}$$

and allows the backward ODE associated to Eq. (4) to be written as (Eq. (A.54))

$$d\boldsymbol{x}_t = -(d \log s_t) \, \boldsymbol{x}_t + \tfrac{1}{2} s_t (d\mathbf{\Sigma}_t) \, \mathbf{\Sigma}_t^{-1} \left[ D(\boldsymbol{x}_t/s_t, \mathbf{\Sigma}_t) - \boldsymbol{x}_t/s_t \right]. \tag{6}$$

This expression suggests our proposed contributions:

- Since $dt$ does not appear in the difference equation, we do not need any strategy to quantify time increments, as needed by all previous approaches. As a consequence, we take the physical time-axis of the time series as the time-axis of the diffusion process.

- Diffusing across the time-series horizon implies that the noise $\mathbf{\Sigma}_t$ and scale $s_t$ schedules acquire physical meanings. Unlike any other diffusion model so far, we empirically estimate these from the data.

The way to estimate these schedules is suggested from their definition. We can show (Eq. (A.5)) that the scale $s_t$ obeys $\mathbb{E}(\boldsymbol{x}_t) = s_t \, \mathbb{E}(\boldsymbol{x}_0)$. Furthermore, we show (Eq. (A.7)) that $\mathrm{Cov}(\boldsymbol{x}_t) = s_t^2 \, \mathbf{\Sigma}_t$.

Therefore, we estimate $s_t$ and $\boldsymbol{\Sigma}_t$ from the input data window $\boldsymbol{y}_{1:T}$, by empirical estimations of the above relations. We follow two approaches:

**Cumulative estimation.** We estimate $s_t$ and $\boldsymbol{\Sigma}_t$ from the cumulative mean and covariance of the data, respectively. That is,

$$\mathbb{E}(\boldsymbol{x}_t) \sim \text{Mean}(\boldsymbol{y}_{1:t}), \quad \mathbb{E}(\boldsymbol{x}_0) \sim \boldsymbol{y}_{1:1}, \quad \Rightarrow \quad \hat{s}_t = \text{Mean}(\boldsymbol{y}_{1:t}) \odot \boldsymbol{y}_{1:1}^{-1}, \tag{7}$$

where the division is element-wise and the reduction is along the sequence axis. Similarly, we estimate $\boldsymbol{\Sigma}_t$ from the cumulative covariance of the data,

$$\text{Cov}(\boldsymbol{x}_t) \sim \text{Cov}(\boldsymbol{y}_{1:t}), \quad \Rightarrow \quad \hat{\boldsymbol{\Sigma}}_t = \hat{S}_t \, \text{Cov}(\boldsymbol{y}_{1:t}) \, \hat{S}_t^T, \tag{8}$$

where $\hat{S}_t = \text{diag}(\hat{s}_{t,1}^{-1}, \ldots, \hat{s}_{t,C}^{-1})$ is a matrix, build from (7), that applies a congruent scaling to the covariance matrix, preserving its positive definiteness.

**Sliding window estimation.** We estimate $s_t$ and $\boldsymbol{\Sigma}_t$ from the mean and covariance of a sliding window over the input $\boldsymbol{y}_{1:T}$. This allows more flexibility to adapt to local changes in the data statistics. It also avoids the problem of defining the variance for the first data point of a window—technically zero, so interpolated in the cumulative estimation above. Additionally, it helps mitigate issues with data values ($\boldsymbol{y}_{1:1}$ in Eq. (7)) close to zero that may blow up the cumulative scale estimate.

## 4.3 TRAINING

As with EDM, we train using denoising score matching. Given subsequences of clean data $\boldsymbol{y} \sim p_{\text{data}}$, we compute the associated subsequences of empirical $\boldsymbol{\Sigma}$. Gaussian noise $\boldsymbol{\varepsilon} \sim \mathcal{N}(\boldsymbol{0}, \boldsymbol{I})$ is drawn and *structured* with this schedule: $\boldsymbol{n} = \boldsymbol{\Sigma}^{1/2} \boldsymbol{\varepsilon}$. Since every time step (and feature) in the data subsequence is corrupted with a different noise level, the noise is no longer i.i.d. as in EDM. The denoiser learns to remove this noise (see Fig. 1a), by minimizing

$$\mathbb{E}_{\boldsymbol{y} \sim p_{\text{data}}, \boldsymbol{\varepsilon} \sim \mathcal{N}(\boldsymbol{0}, \boldsymbol{I})} \left[ \| D_\theta(\boldsymbol{y} + \boldsymbol{n}; \boldsymbol{\Sigma}) - \boldsymbol{y} \|^2 \right]. \tag{9}$$

We evaluate different architectures for the denoiser. They differ in the way the temporal structure is leveraged and whether conditioning on past data is done. We extend the preconditioning scheme of EDM to matrix-valued $\boldsymbol{\Sigma}$. For this, the denoiser is expressed as (Eq. (A.27))

$$D_\theta(\boldsymbol{x}, \boldsymbol{\Sigma}) = \boldsymbol{C}_{\boldsymbol{\Sigma};\text{skip}} \, \boldsymbol{x} + c_{\boldsymbol{\Sigma};\text{out}} \, F_\theta(\boldsymbol{C}_{\boldsymbol{\Sigma};\text{in}} \, \boldsymbol{x}; \boldsymbol{C}_{\boldsymbol{\Sigma};\text{noise}}).$$

By imposing that the inputs and training targets of $F_\theta$ have unit variance, and that its errors are amplified as little as possible, we have (Eqs. (A.35)):

$$\boldsymbol{C}_{\boldsymbol{\Sigma};\text{in}} = (\text{Cov}(\boldsymbol{y}) + \boldsymbol{\Sigma})^{-1/2},$$
$$\boldsymbol{C}_{\boldsymbol{\Sigma};\text{skip}} = \text{Cov}(\boldsymbol{y})(\text{Cov}(\boldsymbol{y}) + \boldsymbol{\Sigma})^{-1},$$
$$c_{\boldsymbol{\Sigma};\text{out}}^2 \boldsymbol{I} = \text{Cov}(\boldsymbol{y})(\text{Cov}(\boldsymbol{y}) + \boldsymbol{\Sigma})^{-1} \boldsymbol{\Sigma},$$
$$\lambda_{\boldsymbol{\Sigma}} = 1/c_{\boldsymbol{\Sigma};\text{out}}^2.$$

The matrix $\boldsymbol{C}_{\boldsymbol{\Sigma};\text{noise}}$ is chosen empirically, as in EDM, and $\lambda_{\boldsymbol{\Sigma}}$ weighs the loss function in Eq. (9). Note that these expressions reduce to those of EDM when $\boldsymbol{\Sigma} = \sigma^2 \boldsymbol{I}$. Furthermore, they hold even when $\boldsymbol{\Sigma}$ is estimated from $\boldsymbol{y}$, provided that the estimator is unbiased.

## 4.4 INFERENCE

Once the denoiser is trained, the score function is estimated by (5). Knowing the score function provides the mechanism for sampling by ODE (or SDE) integration. Since the diffusion and physical time axes are the same, this allows forecasting the next time step by an Euler step of (6) (Eq. (A.55)):

$$\hat{\boldsymbol{y}}_{t+1} = \left[ \boldsymbol{I} - \log \frac{s_t}{s_{t-1}} \, \boldsymbol{I} \right] \hat{\boldsymbol{y}}_t + \tfrac{1}{2} s_t \, (\boldsymbol{\Sigma}_t - \boldsymbol{\Sigma}_{t-1})_+ \, \boldsymbol{\Sigma}_t^{-1} \left[ D_\theta(\hat{\boldsymbol{y}}_t/s_t, \boldsymbol{\Sigma}_t) - \hat{\boldsymbol{y}}_t/s_t \right],$$

where $(\cdot)_+$ denotes the projection onto the cone of positive semi-definite matrices. Here, $s_t$ and $\boldsymbol{\Sigma}_t$ are replaced by their estimates in (7) and (8) (or their sliding window counterparts). This inference

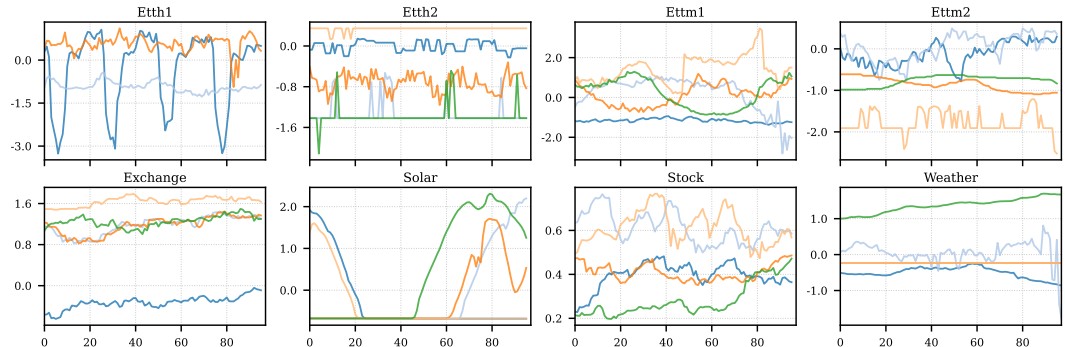

Figure 2: Random test windows from all datasets. Each subplot corresponds to one dataset and shows 5 windows from randomly selected time indices and feature dimensions.

rule is general, and can be implemented by a Cholesky factorization of $\boldsymbol{\Sigma}_t$ that allows applying $\boldsymbol{\Sigma}_t^{-1}$ to vectors. The projection to the positive semi-definite cone requires diagonalization of $\boldsymbol{\Sigma}_t - \boldsymbol{\Sigma}_{t-1}$, which may be costly for large feature size $d$.

An efficient approximation is obtained when the noise covariance is considered diagonal, i.e. $\boldsymbol{\Sigma}_t = \mathrm{diag}(\sigma_{t,1}^2, \ldots, \sigma_{t,C}^2)$. In this case, the inference rule simplifies considerably, since all matrix operations reduce to element-wise operations on the feature dimensions. The Eurler step in this case becomes (Eq. (A.53))

$$\widehat{\boldsymbol{y}}_{t+1} = \left[ \boldsymbol{I} - \log \frac{s_t}{s_{t-1}} \boldsymbol{\Sigma}_t^{1/2} \boldsymbol{\Sigma}_{t-1}^{-1/2} \right] \widehat{\boldsymbol{y}}_t + s_t \left[ \log \boldsymbol{\Sigma}_t^{1/2} \boldsymbol{\Sigma}_{t-1}^{-1/2} \right] D_\theta(\widehat{\boldsymbol{y}}_t / s_t; \boldsymbol{\Sigma}_t). \qquad (10)$$

Remarkably, this inference rule holds exactly beyond the diagonal approximation (see Remark A.1), as long as the principal axes of $\boldsymbol{\Sigma}_t$ do not change with time (only its eigenvalues do). This applies to systems with stable directions of variability but evolving intensities along those directions—e.g. spatial climate patterns, brain signals with stable spatial modes, etc. Due to its simplicity and efficiency, we consider the diagonal approximation for the rest of this work.

Starting from the given data window $\widehat{\boldsymbol{y}}_1 := \boldsymbol{y}_{1:T}$, an Euler step then transforms this window to predict $\widehat{\boldsymbol{y}}_2 := \boldsymbol{y}_{2:T+1}$. Assuming $T = H$ (the horizon length), after $H$ Euler steps (see Fig. 1b), we get the predicted window $\widehat{\boldsymbol{y}}_{1+H} := \boldsymbol{y}_{T+1:T+H}$. Thus inference time is $O(H)$ instead of $O(HS)$ for $S$ diffusion steps, since one Euler step replaces one diffusion step.

Intuitively, every data point in the given window (a single $\boldsymbol{x}_t$ or $\boldsymbol{y}_t$ in our description) may be imagined to be a particle "propagated by diffusion" to a corresponding data point in the predicted window. All these particles in the window are processed in parallel by the network that learned to denoise the signals.

## 5 EXPERIMENTS

### 5.1 DATASETS

We evaluate our approach on eight widely used multivariate datasets: ETTh1, ETTh2, ETTm1, ETTm2, Exchange, Solar, Stock, and Weather. The Electricity-Transformer-Temperature (ETT) datasets (Zhou et al. (2021)) are standard long-sequence forecasting benchmarks, sampled hourly (ETTh) and every 15 minutes (ETTm) from two power-transformer stations. The Exchange dataset (Zhou et al. (2021)) contains daily exchange rates across major currency pairs.

Table 1: Comparison of benchmark time series datasets: data size and feature count.

| Dataset | Timesteps | Dim ($d$) |
|---|---|---|
| ETTh1/ETTh2 | 17,420 | 7 |
| ETTm1/ETTm2 | 69,680 | 7 |
| Exchange | 7,588 | 8 |
| Weather | 52,695 | 21 |
| Solar | 52,560 | 137 |
| Stock | 4,431 | 6 |

The Solar dataset (Lai et al. (2018)) consists of 10-minute solar power measurements from 137 stations, and the Stock dataset provides daily Google stock prices from 2004–2019 (Yoon et al. (2019);

Yuan & Qiao (2024)). Dataset sizes and feature counts are reported in Table 1. Randomly selected test samples are shown in Fig. 2. Further dataset details are provided in Appendix C.1.

## 5.2 DATA PREPROCESSING

It is known that time series forecasting methods requiring extensive preprocessing typically underperform simpler methods (Makridakis et al. (2020)). Such preprocessing include stationarity transformations (Liu et al. (2022)), seasonal adjustments (Findley & Monsell (2019)), multi-resolution analysis (Li et al. (2024a)), etc. These steps often make the approaches cumbersome and introduce inductive biases.

Our method does not require complicated preprocessing. The datasets are partitioned into train (70%), validation (10%), and test (20%) sets, according to the standard practice for these benchmarks (Liu et al. (2024)). Following Gao et al. (2025), we apply z-score normalization to the data. We further partition the sets into subsequences $\boldsymbol{y}_{1:T}$ (windows). For training, each window has $T = H$ (horizon length) time steps, consumed by the denoiser network in batches. For validation and testing, padding is added for TEDM to compute time shifts as appearing in Eq. (10). Each window has $2H$ time steps (plus padding and additional context for conditional denoising). The first $T$ time steps are given to the models for forecasting, the last $H$ steps are the ground truths to contrast with the model predictions. We fix $H = 96$ timesteps for all datasets, as in Gao et al. (2025). See Appendix E for longer horizons.

## 5.3 BASELINES

We are inspired by ARMD (Gao et al. (2025)) and follow their experimental setup closely. However, we include newer diffusion baselines that were not available at the time of their publication. In particular, NsDiff Ye et al. (2025) is a recent approach that extends diffusion models to non-stationary time series, providing a strong baseline for our experiments. They provide a codebase to evaluate TMDM Li et al. (2024b), DiffusionTS Yuan & Qiao (2024), TimeDiff Shen & Kwok (2023), among others. We use their codebase and experimental setup to evaluate these methods for $H = T = 96$.

## 5.4 METRICS

We follow the evaluation methodology of Zhou et al. (2021) and use the mean square error (MSE) and mean absolute error (MAE) to measure performance when comparing z-score normalized data. This has become standard practice in time series forecasting (Gao et al. (2025)). For probabilistic forecasts in Appendix D, we use the standard continuous ranked probability score (CRPS) (Matheson & Winkler (1976)) and quantile interval coverage error (QICE) (Han et al. (2022)).

## 5.5 NETWORK ARCHITECTURES

One of the advantages of elucidating the design space of diffusion models is that the selection of model architectures is more intuitive. The denoiser (as an estimator of the score function used for inference) has a job independent from the forecasting task (see Fig. 1a). This separation makes it ideal for generalization, and gives more flexibility when designing architectures aimed at better denoising outcomes. We considered several architectures, ranging from the simplest Linear network, with $O(T\,d)$ space complexity, to the most complex UNet architecture. These architectures optionally accept a condition on past data, to support conditional denoising (Batzolis et al. (2021)). For more details on these architectures, see Appendix C.3.

## 5.6 MAIN RESULTS

Table 2 shows that TEDM delivers state-of-the-art accuracy across the majority of datasets, achieving the best MSE and MAE on ETTh2 (0.214 / 0.319), ETTm2 (0.135 / 0.253), and Exchange (0.069 / 0.183). On ETTm1, TEDM ranks second (MSE 0.419, MAE 0.421) behind ARMD, and on Weather, TEDM is also second (MSE 0.223, MAE 0.261), close to the strongest baseline on that dataset. The only dataset where TEDM trails the simpler TimeDiff/ARMD pairing is ETTh1 (MSE 0.595, MAE 0.524), which we attribute to typical large-amplitude changes (see Fig. 2) that stress TEDM's assumption of smooth flows (see Assumption A.1). Code is available at `https://gitlab.com/dlr-dw/tedm`.

Table 2: MSE and MAE scores (prediction horizon $H = 96$) for diffusion-based forecasting methods. Best scores per dataset are in **bold**; second best are underlined. Lower is better.

| Methods | Metric | ETTh1 | ETTh2 | ETTm1 | ETTm2 | Exchange | Weather |
|---|---|---|---|---|---|---|---|
| TimeDiff | MSE | **0.417** | 0.364 | 0.548 | 0.209 | 0.208 | 0.228 |
|  | MAE | **0.456** | 0.393 | 0.485 | 0.296 | 0.331 | 0.305 |
| DiffusionTS | MSE | 1.032 | 3.017 | 0.976 | 3.517 | 3.302 | 0.625 |
|  | MAE | 0.757 | 1.340 | 0.726 | 1.472 | 1.493 | 0.609 |
| TMDM | MSE | 0.534 | 0.564 | 0.421 | 0.313 | 0.212 | **0.180** |
|  | MAE | 0.514 | 0.517 | 0.408 | 0.350 | 0.338 | **0.241** |
| ARMD | MSE | 0.445 | 0.311 | **0.337** | 0.181 | 0.093 | 0.232 |
|  | MAE | 0.459 | 0.338 | **0.376** | 0.255 | 0.203 | 0.291 |
| NsDiff | MSE | 0.552 | 0.460 | 0.450 | 0.250 | 0.146 | 0.223 |
|  | MAE | 0.506 | 0.452 | 0.434 | 0.328 | 0.280 | 0.276 |
| **TEDM** | MSE | 0.595 | **0.214** | 0.419 | **0.135** | **0.069** | 0.223 |
|  | MAE | 0.524 | **0.319** | 0.421 | **0.253** | **0.183** | 0.261 |

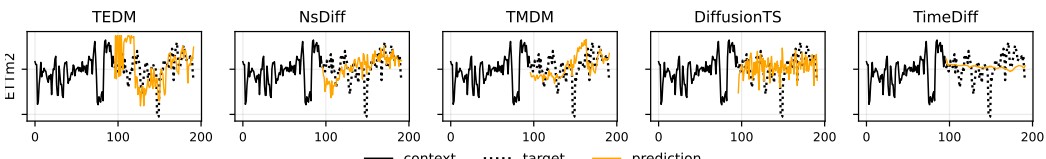

Figure 3: Models evaluated on a sample from the ETTm2 test set.

Table 3: MSE and MAE scores (prediction horizon $H = 96$) for non diffusion-based forecasting methods. Best scores per dataset are in **bold**. Lower is better. Results taken from Gao et al. (2025).

| Methods | Metric | ETTh1 | ETTh2 | ETTm1 | ETTm2 | Exchange | Solar | Stock |
|---|---|---|---|---|---|---|---|---|
| iTransformer | MSE | 0.386 | 0.297 | 0.334 | 0.180 | 0.086 | 0.203 | 0.342 |
|  | MAE | 0.405 | 0.349 | 0.368 | 0.264 | 0.206 | 0.413 | 0.413 |
| TimesNet | MSE | **0.384** | 0.340 | 0.338 | 0.187 | 0.107 | 0.427 | 0.427 |
|  | MAE | 0.402 | 0.347 | 0.375 | 0.267 | 0.234 | 0.499 | 0.499 |
| DLinear | MSE | 0.386 | 0.333 | 0.345 | 0.193 | 0.088 | 0.286 | 0.286 |
|  | MAE | **0.400** | 0.387 | 0.372 | 0.292 | 0.218 | 0.325 | 0.325 |
| PatchTST | MSE | 0.414 | 0.302 | **0.329** | 0.175 | 0.088 | 0.516 | 0.516 |
|  | MAE | 0.419 | 0.348 | **0.367** | 0.259 | 0.205 | 0.524 | 0.524 |
| Client | MSE | 0.392 | 0.305 | 0.336 | 0.184 | 0.086 | 0.352 | 0.352 |
|  | MAE | 0.409 | 0.353 | 0.369 | 0.267 | 0.206 | 0.433 | 0.433 |
| **TEDM** | MSE | 0.595 | **0.214** | 0.419 | **0.135** | **0.069** | 1.061 | **0.056** |
|  | MAE | 0.524 | **0.319** | 0.421 | **0.253** | **0.183** | 0.662 | **0.182** |

These quantitative gains are mirrored by the qualitative behavior in Fig. 3 (more in appendix G): TEDM tracks target trends more faithfully, with better phase alignment and amplitude calibration, and fewer spurious oscillations than competing diffusion models. We also compare with state-of-the-art non-diffusion methods (see Table 3), confirming the superiority of TEDM for several datasets.

These results indicate that elucidating the design space—decoupling the denoiser from the sampler while carefully selecting the schedule and integration strategy—confers consistent advantages in performance across diverse temporal patterns.

Table 4: Ablation study of elucidated models. Lower is better. Percentage gains (in parentheses) indicate improvement of TEDM over EDM.

| Methods | Metric | ETTh2 | ETTm2 | Exchange |
|---------|--------|-------|-------|----------|
| iDDPM+DDIM | MSE | 0.730 | 0.756 | 1.276 |
| | MAE | 0.657 | 0.664 | 0.963 |
| EDM | MSE | 0.419 | 0.293 | 0.448 |
| | MAE | 0.495 | 0.405 | 0.532 |
| TEDM (cumulative $\Sigma_t$, $s_t = 1$) | MSE | 0.303 **(28%)** | 0.137 **(53%)** | 0.110 **(75%)** |
| | MAE | 0.377 **(24%)** | 0.249 **(39%)** | 0.241 **(55%)** |
| TEDM (cumulative $\Sigma_t$, empirical $s_t$) | MSE | 0.242 **(42%)** | 0.135 **(54%)** | 0.068 **(85%)** |
| | MAE | 0.337 **(32%)** | 0.250 **(38%)** | 0.181 **(66%)** |
| TEDM (sliding $\Sigma_t$, empirical $s_t$) | MSE | 0.216 **(49%)** | 0.142 **(52%)** | 0.075 **(83%)** |
| | MAE | 0.259 **(48%)** | 0.249 **(39%)** | 0.195 **(63%)** |

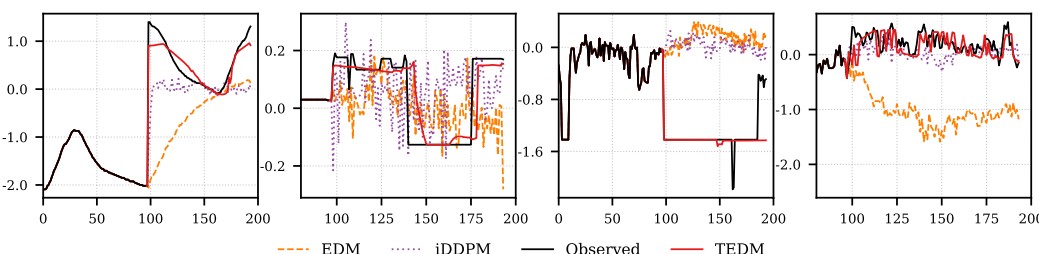

Figure 4: Forecasts generated by EDM, iDDPM+DDIM, and TEDM on four randomly selected subsequences from the ETTm2 dataset. Only a section of the initial subsequences for the right panels is observed for better comparison of the forecasts.

## 5.7 ABLATION STUDIES

We adapt the EDM work of Karras et al. (2022) from images to time series. The inherited modularity allows us to try different noise schedules, time discretization and sampling strategists. For instance, EDM, iDDPM+DDIM, and TEDM, all in a unified codebase. We refer to these as "Elucidated" models, since they fit into a general, modularized diffusion framework, mirroring the terminology from Karras et al. (2022).

Our extension of the EDM framework to time series gives results consistent with those in the vision domain. That is, EDM is consistently better than iDDPM+DDIM, by harnessing optimization of the design space. For time series, this is a result already leveraged by the best weather forecasting framework to date (Price et al. (2025)). Our contribution goes further in two main dimensions:

1. By considering empirical, rather than preset, noise ($\Sigma_t$) and scale ($s_t$) schedules, we get performance gains (see Table 4) of up to **85%** in MSE (**66%** in MAE) with respect to EDM. Fig. 4 shows qualitative results.

2. By aligning the diffusion and physical time axes, we significantly reduce the time complexity of sampling, getting resource benefits comparable to ARMD (see Table 5). The latter does not harness the optimization of the design space though, as shown in the main results.

We also evaluated the loss of skill when considering different forecast horizons. This is shown in Table 6 and compared against the classical skill (Hyndman & Athanasopoulos (2018)) of a forecast that extrapolate the mean of the context window (see appendix B). This is the minimal-skill forecast (yet better than random). TEDM is still able to leverage pattern-wise information to forecast much better than the latest observed average. For a comparison with other methods, see appendix E.

Table 5: Average per-batch training and inference time (seconds), memory (MB), and test MSE on ETTm2. Lower is better.

| Method | Train Time (s) | Train Mem (MB) | Test Time (s) | Test Mem (MB) | MSE |
|---|---|---|---|---|---|
| TimeDiff | 0.022 | 759 | 21.38 | 125 | 0.209 |
| DiffusionTS | 0.098 | 3112 | 634.96 | 14595 | 3.517 |
| TMDM | 0.207 | 15600 | 26.83 | 193 | 0.313 |
| NsDiff | 0.107 | 2682 | 9.80 | 1125 | 0.250 |
| ARMD | 0.009 | **20.7** | **0.02** | **21.3** | 0.181 |
| TEDM | **0.004** | 21.3 | 0.11 | 23.9 | **0.135** |

Table 6: MSE and MAE scores for TEDM on ETTh2, ETTm2, and Exchange for longer forecasting horizons, with baseline (mean) forecast errors. Lower is better.

| Horizon | Metric | ETTh2 | ETTm2 | Exchange | Baseline$_{mean}$ |
|---|---|---|---|---|---|
| 96 | MSE | 0.216 | 0.132 | 0.068 | 1.010 |
| | MAE | 0.321 | 0.251 | 0.182 | 0.801 |
| 192 | MSE | 0.260 | 0.163 | 0.153 | 1.005 |
| | MAE | 0.354 | 0.282 | 0.276 | 0.800 |
| 336 | MSE | 0.326 | 0.248 | 0.283 | 1.003 |
| | MAE | 0.396 | 0.351 | 0.382 | 0.799 |
| 720 | MSE | 0.528 | 0.298 | 0.602 | 1.001 |
| | MAE | 0.510 | 0.386 | 0.571 | 0.798 |

## 6 LIMITATIONS OF THIS WORK

Although the theoretical foundations of TEDM have a general scope within the diffusion framework, not all time series can be represented as the Itô processes underlying such framework. For instance, our formulation cannot capture long-memory dynamics such as those exhibited by fractional Brownian motion (Mandelbrot & Van Ness (1968)). Similarly, our framework cannot represent heavy-tailed or power-law noise (e.g., $\alpha$-stable processes; Samorodnitsky & Taqqu (1994)), nor the jump-driven behaviors (Applebaum (2009)), all of which violate the diffusion regularity assumptions. Furthermore, its effectiveness was shown in the diagonal approximation of the data covariance, which most likely breaks down for datasets with high-dimensional feature space (e.g. Solar in Table 3).

## 7 CONCLUSION AND FUTURE WORK

We present TEDM, the first time series forecasting framework that fully elucidates its design space, grounded on a solid theoretical background. This allows TEDM to reduce the computational complexity to levels suitable for online deployment. We plan to extend our work with a more detailed analysis of the skill in probabilistic forecasting, including a method to sample prediction intervals without ensembling. Aditionally, we foresee the usage of TEDM for anomaly detection, data compression and imputation tasks.

## 8 USE OF LARGE LANGUAGE MODELS

We used large language models (LLMs) only to polish the writing and to help search and organize related work. No modeling ideas, algorithmic designs, experiments, analyses, or reported results were produced by LLMs; all technical content and empirical results were created and verified by the authors.

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

APPENDICES

In the following, we provide theoretical derivations and experimental details of the main results in the paper.

## A  THEORETICAL FOUNDATIONS OF TEDM

### A.1  PRELIMINARY

Following Song et al. (2021), we consider diffusion processes as solutions to an Itô SDE

$$d\boldsymbol{x}_t = \boldsymbol{f}(\boldsymbol{x}_t, t)\, dt + \boldsymbol{G}(\boldsymbol{x}_t, t)\, d\boldsymbol{\omega}_t, \tag{A.1}$$

where $\boldsymbol{f}(\cdot, t) : \mathbb{R}^d \to \mathbb{R}^d$ and $\boldsymbol{G}(\cdot, t) : \mathbb{R}^d \to \mathbb{R}^{d \times d}$. Here, $\boldsymbol{\omega}_t$ is a Wiener process, with changes having zero-mean, $\mathbb{E}(d\boldsymbol{\omega}_t) = 0$, and being uncorrelated: $\mathbb{E}(d\boldsymbol{\omega}_t d\boldsymbol{\omega}_t^T) = \boldsymbol{I} dt$.

To simplify notation, we label functions of time with a subscript, e.g. the drift term is written as $\boldsymbol{f}(\boldsymbol{x}_t, t) = \boldsymbol{f}_t(\boldsymbol{x}_t)$. Also, we explicitly keep the time dependence of $\boldsymbol{x}_t$ when important from the context, otherwise we just use $\boldsymbol{x}$.

We consider the diffusion term, $\boldsymbol{G}(\boldsymbol{x}_t, t) = \boldsymbol{G}_t$, to be independent of $\boldsymbol{x}$ (or slowly varying with $\boldsymbol{x}$). See Assumption A.1 for a more concrete statement about this condition. The SDE then becomes

$$d\boldsymbol{x}_t = \boldsymbol{f}_t(\boldsymbol{x}_t)\, dt + \boldsymbol{G}_t\, d\boldsymbol{\omega}_t. \tag{A.2}$$

In the following, we consider the case of affine drift term, for which the perturbation kernel is Gaussian.

### A.2  PERTURBATION KERNEL

The transition probability density for drift terms of the form $\boldsymbol{f}_t(\boldsymbol{x}) = f_t \boldsymbol{x}$, with $f_t : \mathbb{R} \to \mathbb{R}$, is Gaussian (Eq. 29 in Song et al. (2021)):

$$p_{0t}(\boldsymbol{x}_t | \boldsymbol{x}_0) = \mathcal{N}(\boldsymbol{x}_t; \boldsymbol{\mu}_t, \boldsymbol{V}_t), \tag{A.3}$$

with mean $\boldsymbol{\mu}_t$ and covariance $\boldsymbol{V}_t$. To find these moments, we express (A.2) in the form

$$d\boldsymbol{x}_t = f_t \boldsymbol{x}_t\, dt + \boldsymbol{G}_t\, d\boldsymbol{\omega}_t. \tag{A.4}$$

Taking expectation value

$$\begin{aligned}
d\mathbb{E}(\boldsymbol{x}_t) &= f_t \mathbb{E}(\boldsymbol{x}_t)\, dt + \boldsymbol{G}_t\, \mathbb{E}(d\boldsymbol{\omega}_t) \\
&= f_t \mathbb{E}(\boldsymbol{x}_t)\, dt \\
d\boldsymbol{\mu}_t &= f_t \boldsymbol{\mu}_t dt \\
\frac{d\boldsymbol{\mu}_t}{dt} &= f_t \boldsymbol{\mu}_t \\
\boldsymbol{\mu}_t &= e^{\int_0^t f_\tau d\tau} \boldsymbol{\mu}_0 := s_t \boldsymbol{x}_0,
\end{aligned}$$

where the scale process is defined as

$$\mathbb{E}(\boldsymbol{x}_t) = s_t\, \mathbb{E}(\boldsymbol{x}_0), \qquad s_t = e^{\int_0^t f_\tau d\tau}. \tag{A.5}$$

Using this, and the integral form

$$\boldsymbol{x}_t = s_t \boldsymbol{x}_0 + s_t \int_0^t s_\tau^{-1} \boldsymbol{G}_\tau d\boldsymbol{\omega}_\tau, \tag{A.6}$$

of the Itô process (A.2), we can find the covariance as

$$
\begin{aligned}
\boldsymbol{V}_t &= \mathrm{Cov}(\boldsymbol{x}_t) \\
&= \mathrm{Cov}\left(s_t \int_0^t s_\tau^{-1} \boldsymbol{G}_\tau d\boldsymbol{\omega}_\tau\right) \\
&= s_t^2 \int_0^t s_\tau^{-2} \,\mathrm{Cov}(\boldsymbol{G}_\tau d\boldsymbol{\omega}_\tau) \\
&= s_t^2 \int_0^t s_\tau^{-2} \boldsymbol{G}_\tau \boldsymbol{G}_\tau^T \,\mathrm{Cov}(d\boldsymbol{\omega}_\tau) \\
&= s_t^2 \int_0^t s_\tau^{-2} \boldsymbol{G}_\tau \boldsymbol{G}_\tau^T \boldsymbol{I} d\tau \\
&= s_t^2 \boldsymbol{\Sigma}_t,
\end{aligned}
\tag{A.7}
$$

where we have defined

$$
\boldsymbol{\Sigma}_t := \int_0^t s_\tau^{-2} \boldsymbol{G}_\tau \boldsymbol{G}_\tau^T d\tau \tag{A.8}
$$

$$
s_t^2 \dot{\boldsymbol{\Sigma}}_t = \boldsymbol{G}_t \boldsymbol{G}_t^T. \tag{A.9}
$$

The perturbation kernel (A.3) can then be written as

$$
p_{0t}(\boldsymbol{x}_t|\boldsymbol{x}_0) = \mathcal{N}(\boldsymbol{x}_t; s_t \boldsymbol{x}_0, s_t^2 \boldsymbol{\Sigma}_t). \tag{A.10}
$$

### A.3 SCORE FUNCTION AND DENOISER

The score function is defined as $\nabla_{\boldsymbol{x}} \log p_t(\boldsymbol{x})$, where $p_t(\boldsymbol{x})$ is the marginal distribution. The latter is obtained by integrating (A.10) over all initial conditions:

$$
\begin{aligned}
p_t(\boldsymbol{x}) &= \int_{\mathbb{R}^d} p_{0t}(\boldsymbol{x}|\boldsymbol{x}_0)\, p_{\mathrm{data}}(\boldsymbol{x}_0) d\boldsymbol{x}_0 \\
&= \int_{\mathbb{R}^d} \mathcal{N}(\boldsymbol{x}; s_t \boldsymbol{x}_0, s_t^2 \boldsymbol{\Sigma}_t)\, p_{\mathrm{data}}(\boldsymbol{x}_0) d\boldsymbol{x}_0 \\
&= s_t^{-d} \int_{\mathbb{R}^d} \mathcal{N}(\boldsymbol{x}/s_t; \boldsymbol{x}_0, \boldsymbol{\Sigma}_t)\, p_{\mathrm{data}}(\boldsymbol{x}_0) d\boldsymbol{x}_0 \\
&= s_t^{-d} p(\boldsymbol{x}/s_t, \boldsymbol{\Sigma}_t),
\end{aligned}
\tag{A.11}
$$

where we have used the fact that

$$
\mathcal{N}(\boldsymbol{x}; \boldsymbol{y}, \boldsymbol{\Sigma}) = \frac{1}{(2\pi)^{d/2}|\boldsymbol{\Sigma}|^{1/2}}\, e^{-\frac{1}{2}(\boldsymbol{x}-\boldsymbol{y})^T \boldsymbol{\Sigma}^{-1}(\boldsymbol{x}-\boldsymbol{y})}, \tag{A.12}
$$

and therefore we can express

$$
\mathcal{N}(\boldsymbol{x}_t; s_t \boldsymbol{x}_0, s_t^2 \boldsymbol{\Sigma}_t) = s_t^{-d} \mathcal{N}(\boldsymbol{x}_t/s_t; \boldsymbol{x}_0, \boldsymbol{\Sigma}_t). \tag{A.13}
$$

Following Eq. 19 in Karras et al. (2022), we have defined the mollified version of the data distribution

$$
p(\boldsymbol{x}, \boldsymbol{\Sigma}) = p_{\mathrm{data}}(\boldsymbol{x}) * \mathcal{N}(\boldsymbol{0}; \boldsymbol{\Sigma}), \tag{A.14}
$$

as a convolution that effectively corrupts data samples with Gaussian noise.

**Score function.** The score function is calculated from (A.11)

$$
\begin{aligned}
\nabla_{\boldsymbol{x}} \log p_t(\boldsymbol{x}) &= \nabla_{\boldsymbol{x}} \log\left[s_t^{-d} p(\boldsymbol{x}/s_t, \boldsymbol{\Sigma}_t)\right] \\
&= \nabla_{\boldsymbol{x}} \log p(\boldsymbol{x}/s_t, \boldsymbol{\Sigma}_t) \\
&= \frac{\nabla_{\boldsymbol{x}} p(\boldsymbol{x}/s_t, \boldsymbol{\Sigma}_t)}{p(\boldsymbol{x}/s_t, \boldsymbol{\Sigma}_t)} \\
&= s_t^{-1} \frac{\nabla_{\hat{\boldsymbol{x}}} p(\hat{\boldsymbol{x}}, \boldsymbol{\Sigma}_t)}{p(\hat{\boldsymbol{x}}, \boldsymbol{\Sigma}_t)}, \qquad \hat{\boldsymbol{x}} = \boldsymbol{x}/s_t.
\end{aligned}
\tag{A.15}
$$

To evaluate this, we need an analytical expression for the probability density of the data.

Consider a dataset with a finite number of samples $\{\boldsymbol{y}_1, \cdots, \boldsymbol{y}_N\}$. Assume that these arise from transforming the dataset $\{\boldsymbol{x}_1, \cdots, \boldsymbol{x}_N\}$ using a continuously differentiable mapping $g$, so $Y = g(X)$. The discrete dataset has density $f_X(\boldsymbol{x}) = \sum_{i=1}^N p_{\boldsymbol{x}_i} \delta(\boldsymbol{x} - \boldsymbol{x}_i)$, where $p_{\boldsymbol{x}_i}$ is the probability mass of $\boldsymbol{x}_i$. The density of $Y$ is known to be

$$f_Y(\boldsymbol{y}) = \int_{\mathbb{R}^d} f_X(\boldsymbol{x})\delta(\boldsymbol{y} - g(\boldsymbol{x})) \, d\boldsymbol{x} \tag{A.16}$$

$$= \int_{\mathbb{R}^d} \left[ \sum_{i=1}^N p_{\boldsymbol{x}_i} \delta(\boldsymbol{x} - \boldsymbol{x}_i) \right] \delta(\boldsymbol{y} - g(\boldsymbol{x})) \, d\boldsymbol{x} \tag{A.17}$$

$$= \sum_{i=1}^N p_{\boldsymbol{x}_i} \int_{\mathbb{R}^d} \delta(\boldsymbol{x} - \boldsymbol{x}_i)\delta(\boldsymbol{y} - g(\boldsymbol{x})) \, d\boldsymbol{x} \tag{A.18}$$

$$= \sum_{i=1}^N p_{\boldsymbol{x}_i} \delta(\boldsymbol{y} - g(\boldsymbol{x}_i)) \tag{A.19}$$

$$= \sum_{i=1}^N p_{\boldsymbol{x}_i} \delta(\boldsymbol{y} - \boldsymbol{y}_i). \tag{A.20}$$

In the case of images, $g$ is the identity mapping and $p_{\boldsymbol{x}_i} = 1/N$ is the uniform density. For time-series, one can think of a *propagator* $g$ that takes each $\boldsymbol{x}_i$ into the corresponding $\boldsymbol{y}_i$ after a number of physical time steps. One can then write the data distribution as

$$p_{\text{data}}(\boldsymbol{x}_0) = \sum_{i=1}^N p_{\boldsymbol{x}_i} \delta(\boldsymbol{x}_0 - \boldsymbol{y}_i). \tag{A.21}$$

With this, we can evaluate (A.11) as

$$p(\hat{\boldsymbol{x}}, \boldsymbol{\Sigma}_t) = \int_{\mathbb{R}^d} \mathcal{N}(\hat{\boldsymbol{x}}; \boldsymbol{x}_0, \boldsymbol{\Sigma}_t) \, p_{\text{data}}(\boldsymbol{x}_0) d\boldsymbol{x}_0$$

$$= \int_{\mathbb{R}^d} \mathcal{N}(\hat{\boldsymbol{x}}; \boldsymbol{x}_0, \boldsymbol{\Sigma}_t) \left[ \sum_{i=1}^N p_{\boldsymbol{x}_i} \delta(\boldsymbol{x}_0 - \boldsymbol{y}_i) \right] d\boldsymbol{x}_0$$

$$= \sum_{i=1}^N p_{\boldsymbol{x}_i} \mathcal{N}(\hat{\boldsymbol{x}}; \boldsymbol{y}_i, \boldsymbol{\Sigma}_t) \tag{A.22}$$

$$\nabla_{\hat{\boldsymbol{x}}} \, p(\hat{\boldsymbol{x}}, \boldsymbol{\Sigma}_t) = \sum_{i=1}^N p_{\boldsymbol{x}_i} \nabla_{\hat{\boldsymbol{x}}} \mathcal{N}(\hat{\boldsymbol{x}}; \boldsymbol{y}_i, \boldsymbol{\Sigma}_t) \tag{A.23}$$

From (A.12) we have

$$\nabla_{\hat{\boldsymbol{x}}} \mathcal{N}(\hat{\boldsymbol{x}}; \boldsymbol{y}, \boldsymbol{\Sigma}_t) = \mathcal{N}(\hat{\boldsymbol{x}}; \boldsymbol{y}, \boldsymbol{\Sigma}_t)\nabla_{\hat{\boldsymbol{x}}} \log \mathcal{N}(\hat{\boldsymbol{x}}; \boldsymbol{y}, \boldsymbol{\Sigma}_t)$$

$$= \mathcal{N}(\hat{\boldsymbol{x}}; \boldsymbol{y}, \boldsymbol{\Sigma}_t)\nabla_{\hat{\boldsymbol{x}}} \left[ -\tfrac{1}{2}(\hat{\boldsymbol{x}} - \boldsymbol{y})^T \boldsymbol{\Sigma}_t^{-1}(\hat{\boldsymbol{x}} - \boldsymbol{y}) \right]$$

$$= \mathcal{N}(\hat{\boldsymbol{x}}; \boldsymbol{y}, \boldsymbol{\Sigma}_t) \left[ -\boldsymbol{\Sigma}_t^{-1}(\hat{\boldsymbol{x}} - \boldsymbol{y}) \right],$$

Substituting in (A.23)

$$\nabla_{\hat{\boldsymbol{x}}} \, p(\hat{\boldsymbol{x}}, \boldsymbol{\Sigma}_t) = \sum_{i=1}^N p_{\boldsymbol{x}_i} \mathcal{N}(\hat{\boldsymbol{x}}; \boldsymbol{y}_i, \boldsymbol{\Sigma}_t) \left[ -\boldsymbol{\Sigma}_t^{-1}(\hat{\boldsymbol{x}} - \boldsymbol{y}_i) \right]$$

$$\overset{(A.22)}{=} \boldsymbol{\Sigma}_t^{-1} \left[ p(\hat{\boldsymbol{x}}, \boldsymbol{\Sigma}_t) \, D(\hat{\boldsymbol{x}}, \boldsymbol{\Sigma}_t) - \hat{\boldsymbol{x}} \, p(\hat{\boldsymbol{x}}, \boldsymbol{\Sigma}_t) \right]$$

$$= p(\hat{\boldsymbol{x}}, \boldsymbol{\Sigma}_t) \, \boldsymbol{\Sigma}_t^{-1} \left[ D(\hat{\boldsymbol{x}}, \boldsymbol{\Sigma}_t) - \hat{\boldsymbol{x}} \right]$$

where we have defined

$$D(\hat{\boldsymbol{x}}, \boldsymbol{\Sigma}_t) = \frac{\sum_{i=1}^{N} p_{\boldsymbol{x}_i} \mathcal{N}(\hat{\boldsymbol{x}}; \boldsymbol{y}_i, \boldsymbol{\Sigma}_t) \boldsymbol{y}_i}{\sum_{i=1}^{N} p_{\boldsymbol{x}_i} \mathcal{N}(\hat{\boldsymbol{x}}; \boldsymbol{y}_i, \boldsymbol{\Sigma}_t)}. \tag{A.24}$$

Substituting in (A.15)

$$\nabla_{\boldsymbol{x}} \log p_t(\boldsymbol{x}) = s_t^{-1} \boldsymbol{\Sigma}_t^{-1} \left[ D(\boldsymbol{x}/s_t, \boldsymbol{\Sigma}_t) - \boldsymbol{x}/s_t \right]. \tag{A.25}$$

**Denoiser.** We want to show that (A.25) links the score function to the denoiser. That is, (A.24) is the optimal solution of the denoising objective

$$
\begin{aligned}
\mathcal{L}(D; \boldsymbol{\Sigma}) &= \mathbb{E}_{\boldsymbol{y} \sim p_{\text{data}}} \mathbb{E}_{\boldsymbol{n} \sim \mathcal{N}(\boldsymbol{0}, \boldsymbol{\Sigma})} \| D(\boldsymbol{y} + \boldsymbol{n}, \boldsymbol{\Sigma}) - \boldsymbol{y} \|^2 \\
&= \mathbb{E}_{\boldsymbol{y} \sim p_{\text{data}}} \mathbb{E}_{\boldsymbol{x} \sim \mathcal{N}(\boldsymbol{y}, \boldsymbol{\Sigma})} \| D(\boldsymbol{x}, \boldsymbol{\Sigma}) - \boldsymbol{y} \|^2 \\
&= \mathbb{E}_{\boldsymbol{y} \sim p_{\text{data}}} \int_{\mathbb{R}^d} \mathcal{N}(\boldsymbol{x}; \boldsymbol{y}, \boldsymbol{\Sigma}) \| D(\boldsymbol{x}, \boldsymbol{\Sigma}) - \boldsymbol{y} \|^2 d\boldsymbol{x} \\
&= \int_{\mathbb{R}^d} \sum_{i=1}^{N} p_{\boldsymbol{x}_i} \mathcal{N}(\boldsymbol{x}; \boldsymbol{y}_i, \boldsymbol{\Sigma}) \| D(\boldsymbol{x}, \boldsymbol{\Sigma}) - \boldsymbol{y}_i \|^2 d\boldsymbol{x} \\
&= \int_{\mathbb{R}^d} \mathcal{L}(D; \boldsymbol{x}, \boldsymbol{\Sigma}) \, d\boldsymbol{x}.
\end{aligned}
$$

Since the integrand is positive everywhere, the optimal solution $D_\star$ satisfies

$$D_\star(\boldsymbol{x}, \boldsymbol{\Sigma}) = \arg \min_{D(\boldsymbol{x}, \boldsymbol{\Sigma})} \mathcal{L}(D; \boldsymbol{x}, \boldsymbol{\Sigma}).$$

Since this is a convex optimization problem, the unique solution is found as

$$
\begin{aligned}
\boldsymbol{0} &= \nabla_{D(\boldsymbol{x}, \boldsymbol{\Sigma})} \mathcal{L}(D; \boldsymbol{x}, \boldsymbol{\Sigma}) \\
&= \nabla_{D(\boldsymbol{x}, \boldsymbol{\Sigma})} \sum_{i=1}^{N} p_{\boldsymbol{x}_i} \mathcal{N}(\boldsymbol{x}; \boldsymbol{y}_i, \boldsymbol{\Sigma}) \| D(\boldsymbol{x}, \boldsymbol{\Sigma}) - \boldsymbol{y}_i \|^2 \\
&= \sum_{i=1}^{N} p_{\boldsymbol{x}_i} \mathcal{N}(\boldsymbol{x}; \boldsymbol{y}_i, \boldsymbol{\Sigma}) \nabla_{D(\boldsymbol{x}, \boldsymbol{\Sigma})} \| D(\boldsymbol{x}, \boldsymbol{\Sigma}) - \boldsymbol{y}_i \|^2 \\
&= \sum_{i=1}^{N} p_{\boldsymbol{x}_i} \mathcal{N}(\boldsymbol{x}; \boldsymbol{y}_i, \boldsymbol{\Sigma}) \, 2 \big[ D_\star(\boldsymbol{x}, \boldsymbol{\Sigma}) - \boldsymbol{y}_i \big] \\
D_\star(\boldsymbol{x}, \boldsymbol{\Sigma}) &= \frac{\sum_{i=1}^{N} p_{\boldsymbol{x}_i} \mathcal{N}(\boldsymbol{x}; \boldsymbol{y}_i, \boldsymbol{\Sigma}) \, \boldsymbol{y}_i}{\sum_{i=1}^{N} p_{\boldsymbol{x}_i} \mathcal{N}(\boldsymbol{x}; \boldsymbol{y}_i, \boldsymbol{\Sigma})}.
\end{aligned}
$$

This optimal solution agrees with (A.24).

## A.4 TRAINING WITH PRECONDITIONING

Motivated by Karras et al. (2022), we train the denoiser $D_\theta(\boldsymbol{x}, \boldsymbol{\Sigma})$ by minimizing a weighted version of the denoising objective

$$
\begin{aligned}
\mathcal{L}(D_\theta) &= \mathbb{E}_{\boldsymbol{\Sigma} \sim P_{\boldsymbol{\Sigma}}} \lambda(\boldsymbol{\Sigma}) \, \mathcal{L}(D_\theta; \boldsymbol{\Sigma}), \\
&= \mathbb{E}_{\boldsymbol{\Sigma} \sim P_{\boldsymbol{\Sigma}}} \mathbb{E}_{\boldsymbol{y} \sim p_{\text{data}}} \mathbb{E}_{\boldsymbol{n} \sim \mathcal{N}(\boldsymbol{0}, \boldsymbol{\Sigma})} \lambda(\boldsymbol{\Sigma}) \| D_\theta(\boldsymbol{y} + \boldsymbol{n}, \boldsymbol{\Sigma}) - \boldsymbol{y} \|^2,
\end{aligned} \tag{A.26}
$$

where $P_{\boldsymbol{\Sigma}}$ is a distribution over noise covariances, and $\lambda(\boldsymbol{\Sigma})$ is a weighting function. For simplicity, we write single functions of $\boldsymbol{\Sigma}$ in terms of subscripts, e.g. $\lambda(\boldsymbol{\Sigma}) = \lambda_{\boldsymbol{\Sigma}}$.

We extend the EDM preconditioning strategy as

$$D_\theta(\boldsymbol{x}, \boldsymbol{\Sigma}) = \boldsymbol{C}_{\boldsymbol{\Sigma};\text{skip}} \, \boldsymbol{x} + c_{\boldsymbol{\Sigma};\text{out}} F_\theta(\boldsymbol{C}_{\boldsymbol{\Sigma};\text{in}} \, \boldsymbol{x}; \boldsymbol{C}_{\boldsymbol{\Sigma};\text{noise}}), \tag{A.27}$$

where $F_\theta(\cdot; \cdot)$ is a neural network with parameters $\theta$; $\boldsymbol{C}_{\boldsymbol{\Sigma};\text{skip}}$, $\boldsymbol{C}_{\boldsymbol{\Sigma};\text{in}}$, and $\boldsymbol{C}_{\boldsymbol{\Sigma};\text{noise}}$ are preconditioning matrices that depend on the noise covariance $\boldsymbol{\Sigma}$, and $c_{\boldsymbol{\Sigma};\text{out}}$ is a scalar. Substituting in (A.26)

$$
\begin{aligned}
\mathcal{L}(\theta) &= \mathbb{E}_{\boldsymbol{\Sigma},\boldsymbol{y},\boldsymbol{n}}\, \lambda_{\boldsymbol{\Sigma}} \|\boldsymbol{C}_{\boldsymbol{\Sigma};\text{skip}}(\boldsymbol{y}+\boldsymbol{n}) + c_{\boldsymbol{\Sigma};\text{out}} F_\theta(\boldsymbol{C}_{\boldsymbol{\Sigma};\text{in}}(\boldsymbol{y}+\boldsymbol{n}); \boldsymbol{C}_{\boldsymbol{\Sigma};\text{noise}}) - \boldsymbol{y}\|^2, \\
&= \mathbb{E}_{\boldsymbol{\Sigma},\boldsymbol{y},\boldsymbol{n}}\, \lambda_{\boldsymbol{\Sigma}} c^2_{\boldsymbol{\Sigma};\text{out}} \|F_\theta(\boldsymbol{C}_{\boldsymbol{\Sigma};\text{in}}(\boldsymbol{y}+\boldsymbol{n}); \boldsymbol{C}_{\boldsymbol{\Sigma};\text{noise}}) - \tfrac{1}{c_{\boldsymbol{\Sigma};\text{out}}}(\boldsymbol{y} - \boldsymbol{C}_{\boldsymbol{\Sigma};\text{skip}}(\boldsymbol{y}+\boldsymbol{n}))\|^2, \\
&= \mathbb{E}_{\boldsymbol{\Sigma},\boldsymbol{y},\boldsymbol{n}}\, \lambda_{\boldsymbol{\Sigma}} c^2_{\boldsymbol{\Sigma};\text{out}} \|F_\theta(\boldsymbol{C}_{\boldsymbol{\Sigma};\text{in}}(\boldsymbol{y}+\boldsymbol{n}); \boldsymbol{C}_{\boldsymbol{\Sigma};\text{noise}}) - F_{\text{target}}(\boldsymbol{y}, \boldsymbol{n}, \boldsymbol{\Sigma})\|^2, \quad (A.28)
\end{aligned}
$$

which is just the $l_2$-supervision of the neural network $F_\theta(\cdot; \cdot)$ to match the target function. While doing this, it is required that:

- $F_\theta$ **sees unit-variance inputs.** This is achieved by choosing $\boldsymbol{C}_{\boldsymbol{\Sigma};\text{in}}$ such that

$$
\begin{aligned}
\text{Cov}(\boldsymbol{C}_{\boldsymbol{\Sigma};\text{in}}(\boldsymbol{y}+\boldsymbol{n})) &= \boldsymbol{I}, \\
\boldsymbol{C}_{\boldsymbol{\Sigma};\text{in}}\, \text{Cov}(\boldsymbol{y}+\boldsymbol{n})\, \boldsymbol{C}^T_{\boldsymbol{\Sigma};\text{in}} &= \boldsymbol{I}, \\
\boldsymbol{C}_{\boldsymbol{\Sigma};\text{in}}\, (\text{Cov}(\boldsymbol{y})+\boldsymbol{\Sigma})\, \boldsymbol{C}^T_{\boldsymbol{\Sigma};\text{in}} &= \boldsymbol{I}, \\
\boldsymbol{C}_{\boldsymbol{\Sigma};\text{in}} &= (\text{Cov}(\boldsymbol{y})+\boldsymbol{\Sigma})^{-1/2}. \quad (A.29)
\end{aligned}
$$

  Note that this holds even when $\boldsymbol{\Sigma}$ is estimated from the data $\boldsymbol{y}$, as long as the estimator is unbiased. This is because, in general, from the law of total covariance,

$$
\text{Cov}(\boldsymbol{y}+\boldsymbol{n}) = \mathbb{E}[\text{Cov}(\boldsymbol{y}+\boldsymbol{n}|\boldsymbol{y})] + \text{Cov}(\mathbb{E}[\boldsymbol{y}+\boldsymbol{n}|\boldsymbol{y}]) = \text{Cov}(\boldsymbol{y}) + \mathbb{E}[\hat{\boldsymbol{\Sigma}}],
$$

  since $\text{Cov}(\boldsymbol{y}|\boldsymbol{y}) = 0$, $\text{Cov}(\boldsymbol{n}|\boldsymbol{y}) = \hat{\boldsymbol{\Sigma}}$, and $\mathbb{E}[\boldsymbol{y}+\boldsymbol{n}|\boldsymbol{y}] = \boldsymbol{y}$.

- $F_{\text{target}}$ **gives unit-variance outputs.** This is achieved by choosing $c_{\boldsymbol{\Sigma};\text{out}}$ such that

$$
\begin{aligned}
\text{Cov}(F_{\text{target}}(\boldsymbol{y}, \boldsymbol{n}, \boldsymbol{\Sigma})) &= \boldsymbol{I}, \\
\text{Cov}\left(\tfrac{1}{c_{\boldsymbol{\Sigma};\text{out}}}(\boldsymbol{y} - \boldsymbol{C}_{\boldsymbol{\Sigma};\text{skip}}(\boldsymbol{y}+\boldsymbol{n}))\right) &= \boldsymbol{I}, \\
\tfrac{1}{c^2_{\boldsymbol{\Sigma};\text{out}}} \text{Cov}\left((\boldsymbol{I} - \boldsymbol{C}_{\boldsymbol{\Sigma};\text{skip}})\boldsymbol{y} - \boldsymbol{C}_{\boldsymbol{\Sigma};\text{skip}}\boldsymbol{n}\right) &= \boldsymbol{I}, \\
(\boldsymbol{I} - \boldsymbol{C}_{\boldsymbol{\Sigma};\text{skip}})\, \text{Cov}(\boldsymbol{y})\, (\boldsymbol{I} - \boldsymbol{C}_{\boldsymbol{\Sigma};\text{skip}})^T + \boldsymbol{C}_{\boldsymbol{\Sigma};\text{skip}}\, \boldsymbol{\Sigma}\, \boldsymbol{C}^T_{\boldsymbol{\Sigma};\text{skip}} &= c^2_{\boldsymbol{\Sigma};\text{out}}\boldsymbol{I}. \quad (A.30)
\end{aligned}
$$

- **Errors in $F_\theta$ are amplified as little as possible.** This is achieved by choosing $\boldsymbol{C}_{\boldsymbol{\Sigma};\text{skip}}$ above to minimize $c_{\boldsymbol{\Sigma};\text{out}}$. For this, we can set up the optimization problem

$$
\begin{aligned}
\min_{\boldsymbol{C}_{\boldsymbol{\Sigma};\text{skip}}} \quad & c^2_{\boldsymbol{\Sigma};\text{out}} \\
\text{s.t.} \quad & (\boldsymbol{I} - \boldsymbol{C}_{\boldsymbol{\Sigma};\text{skip}})\, \text{Cov}(\boldsymbol{y})\, (\boldsymbol{I} - \boldsymbol{C}_{\boldsymbol{\Sigma};\text{skip}})^T + \boldsymbol{C}_{\boldsymbol{\Sigma};\text{skip}}\, \boldsymbol{\Sigma}\, \boldsymbol{C}^T_{\boldsymbol{\Sigma};\text{skip}} = c^2_{\boldsymbol{\Sigma};\text{out}}\boldsymbol{I}.
\end{aligned}
$$

  This is a constrained matrix optimization problem that can be solved with Lagrange multipliers. That is, we define the Lagrangian

$$
\begin{aligned}
\mathcal{L}(\boldsymbol{C}_{\boldsymbol{\Sigma};\text{skip}}, c_{\boldsymbol{\Sigma};\text{out}}, \boldsymbol{\Lambda}) = {}& c^2_{\boldsymbol{\Sigma};\text{out}} \\
& + \text{Tr}\Big[\boldsymbol{\Lambda}\Big((\boldsymbol{I} - \boldsymbol{C}_{\boldsymbol{\Sigma};\text{skip}})\, \text{Cov}(\boldsymbol{y})\, (\boldsymbol{I} - \boldsymbol{C}_{\boldsymbol{\Sigma};\text{skip}})^T \\
& \qquad\qquad + \boldsymbol{C}_{\boldsymbol{\Sigma};\text{skip}}\, \boldsymbol{\Sigma}\, \boldsymbol{C}^T_{\boldsymbol{\Sigma};\text{skip}} - c^2_{\boldsymbol{\Sigma};\text{out}}\boldsymbol{I}\Big)\Big].
\end{aligned}
$$

  where $\boldsymbol{\Lambda}$ is a symmetric matrix of Lagrange multipliers. Setting the gradients to zero gives the optimality conditions

$$
\begin{aligned}
\nabla_{\boldsymbol{C}_{\boldsymbol{\Sigma};\text{skip}}} \mathcal{L} &= -2\boldsymbol{\Lambda}(\boldsymbol{I} - \boldsymbol{C}_{\boldsymbol{\Sigma};\text{skip}})\, \text{Cov}(\boldsymbol{y}) + 2\boldsymbol{\Lambda}\, \boldsymbol{C}_{\boldsymbol{\Sigma};\text{skip}}\, \boldsymbol{\Sigma} = \boldsymbol{0}, \\
\nabla_{c_{\boldsymbol{\Sigma};\text{out}}} \mathcal{L} &= 2c_{\boldsymbol{\Sigma};\text{out}} - 2c_{\boldsymbol{\Sigma};\text{out}}\, \text{Tr}(\boldsymbol{\Lambda}) = 0, \\
\nabla_{\boldsymbol{\Lambda}} \mathcal{L} &= (\boldsymbol{I} - \boldsymbol{C}_{\boldsymbol{\Sigma};\text{skip}})\, \text{Cov}(\boldsymbol{y})\, (\boldsymbol{I} - \boldsymbol{C}_{\boldsymbol{\Sigma};\text{skip}})^T + \boldsymbol{C}_{\boldsymbol{\Sigma};\text{skip}}\, \boldsymbol{\Sigma}\, \boldsymbol{C}^T_{\boldsymbol{\Sigma};\text{skip}} - c^2_{\boldsymbol{\Sigma};\text{out}}\boldsymbol{I} \overset{(A.30)}{=} \boldsymbol{0}.
\end{aligned}
$$

  From the second condition, we have that either $c_{\boldsymbol{\Sigma};\text{out}} = 0$ or $\text{Tr}(\boldsymbol{\Lambda}) = 1$. The former is not acceptable, since it would lead to a trivial solution. The latter implies that $\boldsymbol{\Lambda} \neq \boldsymbol{0}$. Therefore,

the first condition can be written as

$$-2(\boldsymbol{I} - \boldsymbol{C}_{\boldsymbol{\Sigma};\text{skip}})\,\text{Cov}(\boldsymbol{y}) + 2\,\boldsymbol{C}_{\boldsymbol{\Sigma};\text{skip}}\,\boldsymbol{\Sigma} = \boldsymbol{0},$$
$$(\boldsymbol{I} - \boldsymbol{C}_{\boldsymbol{\Sigma};\text{skip}})\,\text{Cov}(\boldsymbol{y}) = \boldsymbol{C}_{\boldsymbol{\Sigma};\text{skip}}\,\boldsymbol{\Sigma}, \tag{A.31}$$
$$\boldsymbol{C}_{\boldsymbol{\Sigma};\text{skip}}(\text{Cov}(\boldsymbol{y}) + \boldsymbol{\Sigma}) = \text{Cov}(\boldsymbol{y}),$$
$$\boldsymbol{C}_{\boldsymbol{\Sigma};\text{skip}} = \text{Cov}(\boldsymbol{y})(\text{Cov}(\boldsymbol{y}) + \boldsymbol{\Sigma})^{-1}. \tag{A.32}$$

Substituting (A.31) back in the constraint equation (A.30), we find

$$(\boldsymbol{I} - \boldsymbol{C}_{\boldsymbol{\Sigma};\text{skip}})\,\text{Cov}(\boldsymbol{y})\,(\boldsymbol{I} - \boldsymbol{C}_{\boldsymbol{\Sigma};\text{skip}})^T + \boldsymbol{C}_{\boldsymbol{\Sigma};\text{skip}}\,\boldsymbol{\Sigma}\,\boldsymbol{C}_{\boldsymbol{\Sigma};\text{skip}}^T = c_{\boldsymbol{\Sigma};\text{out}}^2 \boldsymbol{I},$$
$$\boldsymbol{C}_{\boldsymbol{\Sigma};\text{skip}}\,\boldsymbol{\Sigma}\,(\boldsymbol{I} - \boldsymbol{C}_{\boldsymbol{\Sigma};\text{skip}})^T + \boldsymbol{C}_{\boldsymbol{\Sigma};\text{skip}}\,\boldsymbol{\Sigma}\,\boldsymbol{C}_{\boldsymbol{\Sigma};\text{skip}}^T = c_{\boldsymbol{\Sigma};\text{out}}^2 \boldsymbol{I},$$
$$\boldsymbol{C}_{\boldsymbol{\Sigma};\text{skip}}\,\boldsymbol{\Sigma} = c_{\boldsymbol{\Sigma};\text{out}}^2 \boldsymbol{I},$$
$$\text{Cov}(\boldsymbol{y})(\text{Cov}(\boldsymbol{y}) + \boldsymbol{\Sigma})^{-1}\,\boldsymbol{\Sigma} \stackrel{(A.32)}{=} c_{\boldsymbol{\Sigma};\text{out}}^2 \boldsymbol{I}. \tag{A.33}$$

With $\boldsymbol{A}_{\boldsymbol{\Sigma}} = \text{Cov}(\boldsymbol{y})(\text{Cov}(\boldsymbol{y}) + \boldsymbol{\Sigma})^{-1}\,\boldsymbol{\Sigma}$, we see that $\boldsymbol{A}_{\boldsymbol{\Sigma}}\,\boldsymbol{e} = c_{\boldsymbol{\Sigma};\text{out}}^2 \boldsymbol{e}$, so $c_{\boldsymbol{\Sigma};\text{out}}^2$ is the eigenvalue of $\boldsymbol{A}_{\boldsymbol{\Sigma}}$ with eigenvector $\boldsymbol{e}$. To minimize $c_{\boldsymbol{\Sigma};\text{out}}$, it must be chosen as the smallest eigenvalue of $\boldsymbol{A}_{\boldsymbol{\Sigma}}$.

- **The loss weighing is uniform.** This is achieved by choosing $\lambda_{\boldsymbol{\Sigma}}$ such that

$$\lambda_{\boldsymbol{\Sigma}} c_{\boldsymbol{\Sigma};\text{out}}^2 = 1, \tag{A.34}$$

so $\lambda_{\boldsymbol{\Sigma}}$ is the inverse of the smallest eigenvalue of $\boldsymbol{A}_{\boldsymbol{\Sigma}}$.

The preconditioning then ensures that the neural network $F_\theta$ is trained on unit-variance inputs and targets, while minimizing the amplification of errors and ensuring uniform loss weighting.

**Preconditioning summary.** The preconditioning matrices and scalars are

$$\boldsymbol{C}_{\boldsymbol{\Sigma};\text{in}} = (\text{Cov}(\boldsymbol{y}) + \boldsymbol{\Sigma})^{-1/2}, \tag{A.35}$$
$$\boldsymbol{C}_{\boldsymbol{\Sigma};\text{skip}} = \text{Cov}(\boldsymbol{y})(\text{Cov}(\boldsymbol{y}) + \boldsymbol{\Sigma})^{-1}, \tag{A.36}$$
$$c_{\boldsymbol{\Sigma};\text{out}}^2 \boldsymbol{I} = \text{Cov}(\boldsymbol{y})(\text{Cov}(\boldsymbol{y}) + \boldsymbol{\Sigma})^{-1}\,\boldsymbol{\Sigma}, \tag{A.37}$$
$$\lambda_{\boldsymbol{\Sigma}} = 1/c_{\boldsymbol{\Sigma};\text{out}}^2. \tag{A.38}$$

The remaining $\boldsymbol{C}_{\boldsymbol{\Sigma};\text{noise}}$ is chosen empirically, as in EDM.

### A.4.1 SIMPLEST CASE: DIAGONAL $\boldsymbol{\Sigma}$

In the isotropic case, $\boldsymbol{\Sigma} = \sigma^2 \boldsymbol{I}$, the preconditioning reduces to the EDM case, only if the data is assumed identically and independently distributed (i.i.d.), so that $\text{Cov}(\boldsymbol{y}) = \sigma_{\text{data}}^2 \boldsymbol{I}$. In this case, each preconditioning component simplifies to

$$C_{\sigma;\text{in}} = \frac{1}{\sqrt{\sigma_{\text{data}}^2 + \sigma^2}},$$
$$C_{\sigma;\text{skip}} = \frac{\sigma_{\text{data}}^2}{\sigma_{\text{data}}^2 + \sigma^2}$$
$$c_{\sigma;\text{out}}^2 = \frac{\sigma_{\text{data}}^2 \sigma^2}{\sigma_{\text{data}}^2 + \sigma^2},$$
$$\lambda_\sigma = \frac{\sigma_{\text{data}}^2 + \sigma^2}{\sigma_{\text{data}}^2 \sigma^2}.$$

For images, Karras et al. (2022) use $\sigma_{\text{data}} = 0.5$. For standardized time series with unit variance, Price et al. (2025) use $\sigma_{\text{data}} = 1$ for weather forecasting.

In the anisotropic but diagonal case, $\boldsymbol{\Sigma} = \mathrm{diag}(\sigma_1^2, \sigma_2^2, \ldots, \sigma_d^2)$, and assuming i.i.d. data, $\mathrm{Cov}(\boldsymbol{y}) = \sigma_{\mathrm{data}}^2 \boldsymbol{I}$, the preconditioning components become

$$\boldsymbol{C}_{\boldsymbol{\Sigma};\mathrm{in}} = \mathrm{diag}\left( \frac{1}{\sqrt{\sigma_{\mathrm{data}}^2 + \sigma_1^2}}, \frac{1}{\sqrt{\sigma_{\mathrm{data}}^2 + \sigma_2^2}}, \ldots, \frac{1}{\sqrt{\sigma_{\mathrm{data}}^2 + \sigma_d^2}} \right),$$

$$\boldsymbol{C}_{\boldsymbol{\Sigma};\mathrm{skip}} = \mathrm{diag}\left( \frac{\sigma_{\mathrm{data}}^2}{\sigma_{\mathrm{data}}^2 + \sigma_1^2}, \frac{\sigma_{\mathrm{data}}^2}{\sigma_{\mathrm{data}}^2 + \sigma_2^2}, \ldots, \frac{\sigma_{\mathrm{data}}^2}{\sigma_{\mathrm{data}}^2 + \sigma_d^2} \right),$$

$$c_{\boldsymbol{\Sigma};\mathrm{out}}^2 = \min_{j=1,\ldots,d}\left( \frac{\sigma_{\mathrm{data}}^2 \sigma_j^2}{\sigma_{\mathrm{data}}^2 + \sigma_j^2} \right),$$

$$\lambda_{\boldsymbol{\Sigma}} = \frac{1}{c_{\boldsymbol{\Sigma};\mathrm{out}}^2}.$$

This is the next level of complexity, allowing different noise levels per dimension, but still assuming uncorrelated data. This is the case that is used in our experiments, where we set $\sigma_{\mathrm{data}} = 1$, following Price et al. (2025).

### A.4.2 GENERAL CASE: FULL $\boldsymbol{\Sigma}$

In this case, we first notice that the sampled noise $\boldsymbol{n} = \boldsymbol{\Sigma}^{1/2}\boldsymbol{\varepsilon}$, with $\boldsymbol{\varepsilon} \sim \mathcal{N}(\boldsymbol{0}, \boldsymbol{\Sigma})$ is fully structured. But the square root of $\boldsymbol{\Sigma}^{1/2}$ is no longer computed as the element-wise square root of the diagonal elements. For noise sampling, it is sufficient to decompose $\boldsymbol{\Sigma}$ via Cholesky, $\boldsymbol{\Sigma} = \boldsymbol{L}\boldsymbol{L}^T$, where $\boldsymbol{L}$ is a lower-triangular matrix. Then, noise samples are obtained as $\boldsymbol{n} = \boldsymbol{L}\boldsymbol{\varepsilon}$, with $\boldsymbol{\varepsilon} \sim \mathcal{N}(\boldsymbol{0}, \boldsymbol{I})$. Such noise samples will have covariance $\boldsymbol{\Sigma}$, since $\mathrm{Cov}(\boldsymbol{n}) = \boldsymbol{L}\,\mathrm{Cov}(\boldsymbol{\varepsilon})\,\boldsymbol{L}^T = \boldsymbol{L}\boldsymbol{L}^T = \boldsymbol{\Sigma}$.

The preconditioning matrices and scalars can be written, just in terms of $\boldsymbol{\Sigma}$, by using (A.7), so that $\mathrm{Cov}(\boldsymbol{y}) = s^2 \boldsymbol{\Sigma}$. This gives

$$\boldsymbol{C}_{\boldsymbol{\Sigma};\mathrm{in}} = (s^2\boldsymbol{\Sigma} + \boldsymbol{\Sigma})^{-1/2} = \frac{1}{\sqrt{1+s^2}}\boldsymbol{\Sigma}^{-1/2}, \tag{A.39}$$

$$\boldsymbol{C}_{\boldsymbol{\Sigma};\mathrm{skip}} = s^2\boldsymbol{\Sigma}(s^2\boldsymbol{\Sigma} + \boldsymbol{\Sigma})^{-1} = \frac{s^2}{1+s^2}\boldsymbol{I}, \tag{A.40}$$

$$c_{\boldsymbol{\Sigma};\mathrm{out}}^2 = \text{smallest eigenvalue of } \frac{s^2}{1+s^2}\boldsymbol{\Sigma}, \tag{A.41}$$

$$\lambda_{\boldsymbol{\Sigma}} = \frac{1+s^2}{s^2} \cdot \frac{1}{\text{smallest eigenvalue of } \boldsymbol{\Sigma}}. \tag{A.42}$$

Note that $\boldsymbol{C}_{\boldsymbol{\Sigma};\mathrm{in}}$ can be efficiently applied to vectors $\boldsymbol{v}$ only in terms of $\boldsymbol{L}^{-1}\boldsymbol{v}$ (up to an orthogonal rotation). This is due to the known relation $\boldsymbol{\Sigma}^{-1/2}\boldsymbol{v} = \boldsymbol{Q}\boldsymbol{L}^{-1}\boldsymbol{v}$, for some orthogonal matrix $\boldsymbol{Q}$. Thus, we only need to solve linear systems with $\boldsymbol{L}$, which is efficient since $\boldsymbol{L}$ is lower-triangular.

### A.5 SAMPLING DURING INFERENCE

The general forward SDE (A.1)

$$d\boldsymbol{x} = \boldsymbol{f}_t(\boldsymbol{x})\,dt + \boldsymbol{G}_t(\boldsymbol{x})\,d\boldsymbol{\omega}_t, \tag{A.43}$$

has no infomation about the data distribution $p_{\mathrm{data}}(\boldsymbol{x})$. Such information is learned through the score function $\nabla_{\boldsymbol{x}}\log p_t(\boldsymbol{x})$, where $p_t(\boldsymbol{x})$ is the marginal density of $\boldsymbol{x}_t$ at time $t$. There are two main ways of incorporating this information into the sampling process:

1. Deterministic (ODE): by removing the noise term from the SDE, and adjusting the drift term to include the score function.
2. Stochastic (SDE): by going backwards in time with a backward SDE that includes the score function in the drift term.

Both ways start from the forward Kolmogorov (or Fokker-Planck) equation, which describes how the marginal density $p_t(\boldsymbol{x})$ evolves with time. We write it as a continuity equation, by defining the

probability flux

$$\boldsymbol{J}_t(\boldsymbol{x}) = \boldsymbol{f}_t(\boldsymbol{x})\, p_t(\boldsymbol{x}) - \tfrac{1}{2}\nabla_{\boldsymbol{x}} \cdot \left[\boldsymbol{G}_t(\boldsymbol{x})\boldsymbol{G}_t^T(\boldsymbol{x})p_t(\boldsymbol{x})\right]. \tag{A.44}$$

With this, density changes occur by flux transport

$$\frac{\partial p_t(\boldsymbol{x})}{\partial t} = -\nabla_{\boldsymbol{x}} \cdot \boldsymbol{J}_t(\boldsymbol{x}). \tag{A.45}$$

**Assumption A.1** (Smooth flows). The changes $\nabla_{\boldsymbol{x}} \cdot \left[\boldsymbol{G}_t(\boldsymbol{x})\boldsymbol{G}_t^T(\boldsymbol{x})\right]$ are negligible. That is, there exist $c > 0$ such that $\|\nabla_{\boldsymbol{x}} \cdot \left[\boldsymbol{G}_t(\boldsymbol{x})\boldsymbol{G}_t^T(\boldsymbol{x})\right]\| \ll c\,\|\boldsymbol{G}_t(\boldsymbol{x})\boldsymbol{G}_t^T(\boldsymbol{x})\|$. This allows to drop the $\boldsymbol{x}$-dependence of $\boldsymbol{G}_t(\boldsymbol{x})\boldsymbol{G}_t^T(\boldsymbol{x})$ and, from (A.9), just write $\boldsymbol{G}_t\boldsymbol{G}_t^T = s_t^2\dot{\boldsymbol{\Sigma}}_t$

Intuitively, the temporal rate of $\boldsymbol{\Sigma}_t$ does not vary considerably from sample $\boldsymbol{x}$ to sample. This leaves jump processes out of scope. For these, Itô processes have to be generalized to jump-diffusion stochastic dynamics Anvari et al. (2016).

Using assumption A.1, we can write the flux (A.44) as

$$\begin{aligned} \boldsymbol{J}_t(\boldsymbol{x}) &= \boldsymbol{f}_t(\boldsymbol{x})\, p_t(\boldsymbol{x}) - \tfrac{1}{2}\boldsymbol{G}_t\boldsymbol{G}_t^T\nabla_{\boldsymbol{x}}p_t(\boldsymbol{x}), \\ &= \boldsymbol{f}_t(\boldsymbol{x})\, p_t(\boldsymbol{x}) - \tfrac{1}{2}s_t^2\dot{\boldsymbol{\Sigma}}_t\nabla_{\boldsymbol{x}}p_t(\boldsymbol{x}). \end{aligned} \tag{A.46}$$

### A.5.1 DETERMINISTIC SAMPLING

This is achieved by obtaining values of $\boldsymbol{x}_t$ without the noise term in the SDE, but still distributed according to $p_t(\boldsymbol{x})$. We see from (A.43) that making the diffusion term $\boldsymbol{G}_t(\boldsymbol{x}) = \boldsymbol{0}$ removes the noise from the SDE. This manifests, from (A.44), as the probability flux being proportional to the drift term $\boldsymbol{f}_t(\boldsymbol{x})$. This gives a general recipe for obtaining the desired ODE: find a new process

$$d\boldsymbol{x} = \boldsymbol{f}_t^*(\boldsymbol{x})\, dt, \tag{A.47}$$

that has the same probability flux (A.46) as the original, but proportional to the drift term $\boldsymbol{f}_t^*(\boldsymbol{x})$. We can rewrite (A.46) as

$$\begin{aligned} \boldsymbol{J}_t(\boldsymbol{x}) &= \boldsymbol{f}_t(\boldsymbol{x})\, p_t(\boldsymbol{x}) - \tfrac{1}{2}s_t^2\dot{\boldsymbol{\Sigma}}_t\nabla_{\boldsymbol{x}}p_t(\boldsymbol{x}) \\ &= \boldsymbol{f}_t(\boldsymbol{x})\, p_t(\boldsymbol{x}) - \tfrac{1}{2}s_t^2\dot{\boldsymbol{\Sigma}}_t\, p_t(\boldsymbol{x})\, \nabla_{\boldsymbol{x}}\log p_t(\boldsymbol{x}) \\ &= \left[\boldsymbol{f}_t(\boldsymbol{x}) - \tfrac{1}{2}s_t^2\dot{\boldsymbol{\Sigma}}_t\, \nabla_{\boldsymbol{x}}\log p_t(\boldsymbol{x})\right] p_t(\boldsymbol{x}) \\ &= \boldsymbol{f}_t^*(\boldsymbol{x})\, p_t(\boldsymbol{x}), \end{aligned}$$

from which the new drift term $\boldsymbol{f}_t^*(\boldsymbol{x})$ is readily obtained. The ODE running backward in time is obtained, from (A.47), by changing the sign of $\boldsymbol{f}_t^*(\boldsymbol{x})$. This is the one used for deterministic sampling, in which backward evolution is linked to denoising:

$$\frac{d\boldsymbol{x}}{dt} = -\boldsymbol{f}_t(\boldsymbol{x}) + \tfrac{1}{2}s_t^2\dot{\boldsymbol{\Sigma}}_t\, \nabla_{\boldsymbol{x}}\log p_t(\boldsymbol{x}). \tag{A.48}$$

Bringing back the affine drift $\boldsymbol{f}_t(\boldsymbol{x}) = f_t\boldsymbol{x} \stackrel{(A.5)}{=} (\dot{s}_t/s_t)\,\boldsymbol{x}$, and the score from (A.25), we have that

$$\begin{aligned} \frac{d\boldsymbol{x}}{dt} &= -\frac{\dot{s}_t}{s_t}\boldsymbol{x} + \tfrac{1}{2}s_t^2\dot{\boldsymbol{\Sigma}}_t\, s_t^{-1}\, \boldsymbol{\Sigma}_t^{-1}\left[D(\boldsymbol{x}/s_t, \boldsymbol{\Sigma}_t) - \boldsymbol{x}/s_t\right], \\ &= -\frac{d\log s_t}{dt}\boldsymbol{x} + \tfrac{1}{2}s_t\dot{\boldsymbol{\Sigma}}_t\, \boldsymbol{\Sigma}_t^{-1}\left[D(\boldsymbol{x}/s_t, \boldsymbol{\Sigma}_t) - \boldsymbol{x}/s_t\right] \end{aligned} \tag{A.49}$$

**Simplest cases: $\boldsymbol{\Sigma}_t$ commutes with $\dot{\boldsymbol{\Sigma}}_t$.** We want to find the conditions under which the ODE (A.49) can be expressed in terms of logarithmic differentials of both $s_t$ and $\boldsymbol{\Sigma}_t$.

**Lemma A.1.** If $\boldsymbol{\Sigma}_t$ commutes with $\dot{\boldsymbol{\Sigma}}_t$, then

$$\frac{d\log\boldsymbol{\Sigma}_t}{dt} = \dot{\boldsymbol{\Sigma}}_t\, \boldsymbol{\Sigma}_t^{-1}. \tag{A.50}$$

*Proof.* The Daleckii-Krein formula Higham (2008), for $\mathbf{\Gamma}_t = \log \mathbf{\Sigma}_t$, reads

$$\dot{\mathbf{\Gamma}}_t = \int_0^\infty (\mathbf{\Sigma}_t + \eta \mathbf{I})^{-1} \dot{\mathbf{\Sigma}}_t (\mathbf{\Sigma}_t + \eta \mathbf{I})^{-1} d\eta. \tag{A.51}$$

If $\mathbf{\Sigma}_t$ commutes with $\dot{\mathbf{\Sigma}}_t$ (i.e. $[\mathbf{\Sigma}_t, \dot{\mathbf{\Sigma}}_t] = \mathbf{0}$), then they can be diagonalized simultaneously, so that

$$\begin{aligned}
\dot{\mathbf{\Gamma}}_t &= \int_0^\infty \left(\mathbf{U}\mathbf{\Lambda}_t\mathbf{U}^T + \eta\mathbf{I}\right)^{-1} \mathbf{U}\dot{\mathbf{\Lambda}}_t\mathbf{U}^T \left(\mathbf{U}\mathbf{\Lambda}_t\mathbf{U}^T + \eta\mathbf{I}\right)^{-1} d\eta, \\
&= \mathbf{U}\left(\int_0^\infty (\mathbf{\Lambda}_t + \eta\mathbf{I})^{-1} \dot{\mathbf{\Lambda}}_t (\mathbf{\Lambda}_t + \eta\mathbf{I})^{-1} d\eta\right) \mathbf{U}^T, \\
&= \mathbf{U}\left(\int_0^\infty \operatorname{diag}\left(\frac{\dot{\lambda}_{t,1}}{(\lambda_{t,1} + \eta)^2}, \frac{\dot{\lambda}_{t,2}}{(\lambda_{t,2} + \eta)^2}, \ldots, \frac{\dot{\lambda}_{t,d}}{(\lambda_{t,d} + \eta)^2}\right) d\eta\right) \mathbf{U}^T, \\
&= \mathbf{U}\operatorname{diag}\left(\frac{\dot{\lambda}_{t,1}}{\lambda_{t,1}}, \frac{\dot{\lambda}_{t,2}}{\lambda_{t,2}}, \ldots, \frac{\dot{\lambda}_{t,d}}{\lambda_{t,d}}\right) \mathbf{U}^T, \\
&= \mathbf{U}\dot{\mathbf{\Lambda}}_t \mathbf{\Lambda}_t^{-1}\mathbf{U}^T, \\
&= \dot{\mathbf{\Sigma}}_t \mathbf{\Sigma}_t^{-1},
\end{aligned}$$

where the $\eta$-integrals were computed element-wise, as $\int_0^\infty (\lambda + \eta)^{-2} d\eta = 1/\lambda$. ∎

Under such a commutation condition, we can rewrite (A.49) as

$$\begin{aligned}
\frac{d\mathbf{x}}{dt} &= -\frac{d\log s_t}{dt}\mathbf{x} + \tfrac{1}{2}s_t\frac{d\log\mathbf{\Sigma}_t}{dt}\left[D(\mathbf{x}/s_t, \mathbf{\Sigma}_t) - \mathbf{x}/s_t\right] \\
d\mathbf{x} &= -\left[d\log s_t + \tfrac{1}{2}d\log\mathbf{\Sigma}_t\right]\mathbf{x} + \tfrac{1}{2}s_t\left[d\log\mathbf{\Sigma}_t\right]D(\mathbf{x}/s_t, \mathbf{\Sigma}_t) \\
&= -\left[d\log(s_t\mathbf{\Sigma}_t^{1/2})\right]\mathbf{x} + s_t\left[d\log\mathbf{\Sigma}_t^{1/2}\right]D(\mathbf{x}/s_t, \mathbf{\Sigma}_t),
\end{aligned}$$

where we have used the property $\log\mathbf{\Sigma}^k = k\log\mathbf{\Sigma}$, always valid for symmetric positive definite matrices, and $\log(a\mathbf{\Sigma}) = \log(a)\mathbf{I} + \log\mathbf{\Sigma}$, for scalar $a$. The latter is valid for any matrix $\mathbf{\Sigma}$.

This gives the deterministic ODE:

$$d\mathbf{x}_t = -\left[d\log(s_t\mathbf{\Sigma}_t^{1/2})\right]\mathbf{x}_t + s_t\left[d\log\mathbf{\Sigma}_t^{1/2}\right]D(\mathbf{x}_t/s_t, \mathbf{\Sigma}_t). \tag{A.52}$$

Now, in an Euler step, we can approximate $[\mathbf{\Sigma}_t, \dot{\mathbf{\Sigma}}_t] = \mathbf{0} = [\mathbf{\Sigma}_t, \mathbf{\Sigma}_t - \mathbf{\Sigma}_{t-1}] = -[\mathbf{\Sigma}_t, \mathbf{\Sigma}_{t-1}]$. Therefore, given functions $f$ and $g$, $\log[f(\mathbf{\Sigma}_t)g(\mathbf{\Sigma}_{t-1})] = \log f(\mathbf{\Sigma}_t) + \log g(\mathbf{\Sigma}_{t-1})$. This can be used to write an Euler step of (A.52) as

$$\begin{aligned}
\mathbf{x}_{t+1} - \mathbf{x}_t &= -\left[\log(s_t\mathbf{\Sigma}_t^{1/2}) - \log(s_{t-1}\mathbf{\Sigma}_{t-1}^{1/2})\right]\mathbf{x}_t + s_t\left[\log\mathbf{\Sigma}_t^{1/2} - \log\mathbf{\Sigma}_{t-1}^{1/2}\right]D(\mathbf{x}_t/s_t, \mathbf{\Sigma}_t) \\
&= -\left[\mathbf{I}\log\frac{s_t}{s_{t-1}} + \left(\log\mathbf{\Sigma}_t^{1/2} - \log\mathbf{\Sigma}_{t-1}^{1/2}\right)\right]\mathbf{x}_t + s_t\left[\log\mathbf{\Sigma}_t^{1/2} - \log\mathbf{\Sigma}_{t-1}^{1/2}\right]D(\mathbf{x}_t/s_t, \mathbf{\Sigma}_t) \\
&= -\left[\log\frac{s_t}{s_{t-1}}\mathbf{\Sigma}_t^{1/2}\mathbf{\Sigma}_{t-1}^{-1/2}\right]\mathbf{x}_t + s_t\left[\log\mathbf{\Sigma}_t^{1/2}\mathbf{\Sigma}_{t-1}^{-1/2}\right]D(\mathbf{x}_t/s_t, \mathbf{\Sigma}_t) \\
\mathbf{x}_{t+1} &= \left[\mathbf{I} - \log\frac{s_t}{s_{t-1}}\mathbf{\Sigma}_t^{1/2}\mathbf{\Sigma}_{t-1}^{-1/2}\right]\mathbf{x}_t + s_t\left[\log\mathbf{\Sigma}_t^{1/2}\mathbf{\Sigma}_{t-1}^{-1/2}\right]D(\mathbf{x}_t/s_t, \mathbf{\Sigma}_t), \tag{A.53}
\end{aligned}$$

where we have approximated the differentials of the logarithms as backward differences that exploit the current and previous steps.

*Remark* A.1. The case of $\mathbf{\Sigma}_t$ being diagonal (considered in the main text) is included in the commutation condition, since diagonal matrices always commute. However, the inference formula of this section applies more generally to processes for which the principal axes of $\mathbf{\Sigma}_t$ remain fixed in time, while only the eigenvalues change. This was assumed when writing $\mathbf{\Sigma}_t = \mathbf{U}\mathbf{\Lambda}_t\mathbf{U}^T$, with fixed orthogonal $\mathbf{U}$ and time-varying diagonal $\mathbf{\Lambda}_t$.

**General case: unconstrained $\Sigma_t$.** We can have processes respecting assumption A.1, with the principal axes of $\Sigma_t$ allowed to change with time. Deterministic sampling in this case, can be obtained from (A.49), which can be written as

$$
\begin{aligned}
\frac{d\boldsymbol{x}_t}{dt} &= -\frac{d\log s_t}{dt}\boldsymbol{x}_t + \tfrac{1}{2}s_t\dot{\boldsymbol{\Sigma}}_t\,\boldsymbol{\Sigma}_t^{-1}\left[D(\boldsymbol{x}_t/s_t,\boldsymbol{\Sigma}_t) - \boldsymbol{x}_t/s_t\right], \\
&= -\frac{d\log s_t}{dt}\boldsymbol{x}_t + \tfrac{1}{2}s_t\dot{\boldsymbol{\Sigma}}_t\,\boldsymbol{\Sigma}_t^{-1}\left[D(\boldsymbol{x}_t/s_t,\boldsymbol{\Sigma}_t) - \boldsymbol{x}_t/s_t\right], \\
d\boldsymbol{x}_t &= -d\log s_t\,\boldsymbol{x}_t + \tfrac{1}{2}s_t d\boldsymbol{\Sigma}_t\,\boldsymbol{\Sigma}_t^{-1}\left[D(\boldsymbol{x}_t/s_t,\boldsymbol{\Sigma}_t) - \boldsymbol{x}_t/s_t\right].
\end{aligned}
\tag{A.54}
$$

An Euler step of this reads,

$$
\boldsymbol{x}_{t+1} - \boldsymbol{x}_t = -\left(\log s_t - \log s_{t-1}\right)\boldsymbol{x}_t + \tfrac{1}{2}s_t\left(\boldsymbol{\Sigma}_t - \boldsymbol{\Sigma}_{t-1}\right)_+\boldsymbol{\Sigma}_t^{-1}\left[D(\boldsymbol{x}_t/s_t,\boldsymbol{\Sigma}_t) - \boldsymbol{x}_t/s_t\right],
$$

$$
\boldsymbol{x}_{t+1} = \left[\boldsymbol{I} - \log\frac{s_t}{s_{t-1}}\boldsymbol{I}\right]\boldsymbol{x}_t + \tfrac{1}{2}s_t\left(\boldsymbol{\Sigma}_t - \boldsymbol{\Sigma}_{t-1}\right)_+\boldsymbol{\Sigma}_t^{-1}\left[D(\boldsymbol{x}_t/s_t,\boldsymbol{\Sigma}_t) - \boldsymbol{x}_t/s_t\right], \tag{A.55}
$$

where $(\cdot)_+$ denotes the projection onto the cone of positive semi-definite matrices—since $\dot{\boldsymbol{\Sigma}}_t \succ \boldsymbol{0}$ for the Itô diffusion to be well defined, $d\boldsymbol{\Sigma}_t \succ \boldsymbol{0}$ as well, and hence its finite-difference approximations. Again, as in the general preconditioning case, the matrix $\boldsymbol{\Sigma}_t^{-1}$ has to be applied to vectors via $\boldsymbol{L}_t^{-1}$, where $\boldsymbol{\Sigma}_t = \boldsymbol{L}_t\boldsymbol{L}_t^T$ is the Cholesky decomposition of $\boldsymbol{\Sigma}_t$.

### A.5.2 STOCHASTIC SAMPLING

We could also sample from the data distribution by going backwards in time with a backward SDE. We anticipated how to do this with the ODE, by reversing the sign of the flux term. This manifested itself in the sign change of the drift term in (A.47).

In general, the time reversal entails a new Itô SDE of the form

$$
d\boldsymbol{x} = \tilde{\boldsymbol{f}}_t(\boldsymbol{x})\,dt + \boldsymbol{G}_t(\boldsymbol{x})\,d\tilde{\boldsymbol{\omega}}_t, \tag{A.56}
$$

where the probability flux is reversed

$$
\tilde{\boldsymbol{J}}_t(\boldsymbol{x}) = -\boldsymbol{J}_t(\boldsymbol{x}). \tag{A.57}
$$

Writing (A.46) in a form proportional to $p_t(\boldsymbol{x})$, we have $\boldsymbol{J}_t = \left[\boldsymbol{f}_t(\boldsymbol{x}) - \tfrac{1}{2}s_t^2\dot{\boldsymbol{\Sigma}}_t\,\nabla_{\boldsymbol{x}}\log p_t(\boldsymbol{x})\right]p_t(\boldsymbol{x})$. Therefore, from (A.57), we get

$$
\left[\tilde{\boldsymbol{f}}_t(\boldsymbol{x}) - \tfrac{1}{2}s_t^2\dot{\boldsymbol{\Sigma}}_t\,\nabla_{\boldsymbol{x}}\log p_t(\boldsymbol{x})\right]p_t(\boldsymbol{x}) = -\left[\boldsymbol{f}_t(\boldsymbol{x}) - \tfrac{1}{2}s_t^2\dot{\boldsymbol{\Sigma}}_t\,\nabla_{\boldsymbol{x}}\log p_t(\boldsymbol{x})\right]p_t(\boldsymbol{x})
$$

$$
\tilde{\boldsymbol{f}}_t(\boldsymbol{x}) - \tfrac{1}{2}s_t^2\dot{\boldsymbol{\Sigma}}_t\,\nabla_{\boldsymbol{x}}\log p_t(\boldsymbol{x}) = -\left[\boldsymbol{f}_t(\boldsymbol{x}) - \tfrac{1}{2}s_t^2\dot{\boldsymbol{\Sigma}}_t\,\nabla_{\boldsymbol{x}}\log p_t(\boldsymbol{x})\right]
$$

$$
\tilde{\boldsymbol{f}}_t(\boldsymbol{x}) = -\boldsymbol{f}_t(\boldsymbol{x}) + s_t^2\dot{\boldsymbol{\Sigma}}_t\,\nabla_{\boldsymbol{x}}\log p_t(\boldsymbol{x})
$$

The backward SDE (A.56) thus acquires the form

$$
d\boldsymbol{x} = \left[-\boldsymbol{f}_t(\boldsymbol{x}) + s_t^2\dot{\boldsymbol{\Sigma}}_t\,\nabla_{\boldsymbol{x}}\log p_t(\boldsymbol{x})\right]dt + \boldsymbol{G}_t\,d\tilde{\boldsymbol{\omega}}_t, \tag{A.58}
$$

containing the score function in the new drift term.

Karras et al. (2022) derived a SDE sampler for isotropic diffusion. Here, we extend their derivation to the anisotropic but diagonal case.

**Simplest case: Diagonal $\Sigma_t$.** We consider the anisotropic heat equation

$$
\frac{\partial q_t(\boldsymbol{x})}{\partial t} = \nabla_{\boldsymbol{x}} \cdot \boldsymbol{K}_t\nabla_{\boldsymbol{x}}q_t(\boldsymbol{x}), \tag{A.59}
$$

whose solution, with initial value $q_0(\boldsymbol{x}) := p_{\text{data}}(\boldsymbol{x})$, is the marginal density $q_t(\boldsymbol{x}) = p_t(\boldsymbol{x})$. The matrix $\boldsymbol{K}_t$ is considered diagonal, with different elements along the diagonal implying anisotropy. Taking Fourier transform along the $\boldsymbol{x}$-dimension, we get

$$\frac{\partial \hat{q}_t(\boldsymbol{\nu})}{\partial t} = - \left( \boldsymbol{\nu}^T \boldsymbol{K}_t \, \boldsymbol{\nu} \right) \hat{q}_t(\boldsymbol{\nu}), \tag{A.60}$$

The target solution $q_t(\boldsymbol{x}) = p_t(\boldsymbol{x})$ and its Fourier transform $\hat{q}_t(\boldsymbol{\nu})$ are given by (A.11) and (A.14)

$$q_t(\boldsymbol{x}) = s_t^{-d} p_{\text{data}}(\boldsymbol{x}/s_t) * \mathcal{N}(\boldsymbol{0}; \boldsymbol{\Sigma}_t) \tag{A.61}$$

$$\hat{q}_t(\boldsymbol{\nu}) = \hat{p}_{\text{data}}(\boldsymbol{\nu}) \, \exp \left[ -\tfrac{1}{2} \boldsymbol{\nu}^T \boldsymbol{\Sigma}_t \, \boldsymbol{\nu} \right] . \tag{A.62}$$

Differentiating (A.62) along the time axis, we have

$$\frac{\partial \hat{q}_t(\boldsymbol{\nu})}{\partial t} = - \left( \tfrac{1}{2} \boldsymbol{\nu}^T \dot{\boldsymbol{\Sigma}}_t \, \boldsymbol{\nu} \right) \hat{q}_t(\boldsymbol{\nu}). \tag{A.63}$$

Equating with the right-hand side of (A.60), we get

$$\boldsymbol{\nu}^T \boldsymbol{K}_t \, \boldsymbol{\nu} = \tfrac{1}{2} \boldsymbol{\nu}^T \dot{\boldsymbol{\Sigma}}_t \, \boldsymbol{\nu}$$

$$\boldsymbol{K}_t = \tfrac{1}{2} \dot{\boldsymbol{\Sigma}}_t,$$

the second equality resulting from assuming $\boldsymbol{\Sigma}_t$ diagonal. Substituting in (A.59) we have

$$\frac{\partial p_t(\boldsymbol{x})}{\partial t} = \tfrac{1}{2} \nabla_{\boldsymbol{x}} \cdot \dot{\boldsymbol{\Sigma}}_t \nabla_{\boldsymbol{x}} p_t(\boldsymbol{x}), \tag{A.64}$$

$$\overset{(A.45)}{=} -\nabla_{\boldsymbol{x}} \cdot \boldsymbol{J}_t(\boldsymbol{x}).$$

Equating the right-hand sides of (A.64) and (A.45), we get

$$-\boldsymbol{J}_t(\boldsymbol{x}) = \tfrac{1}{2} \dot{\boldsymbol{\Sigma}}_t \nabla_{\boldsymbol{x}} p_t(\boldsymbol{x})$$

$$-\boldsymbol{f}_t(\boldsymbol{x}) \, p_t(\boldsymbol{x}) + \tfrac{1}{2} s_t^2 \dot{\boldsymbol{\Sigma}}_t \nabla_{\boldsymbol{x}} p_t(\boldsymbol{x}) = \tfrac{1}{2} \dot{\boldsymbol{\Sigma}}_t \nabla_{\boldsymbol{x}} p_t(\boldsymbol{x})$$

$$-\boldsymbol{f}_t(\boldsymbol{x}) \, p_t(\boldsymbol{x}) = \tfrac{1}{2} (1 - s_t^2) \dot{\boldsymbol{\Sigma}}_t \nabla_{\boldsymbol{x}} p_t(\boldsymbol{x})$$

$$\boldsymbol{f}_t(\boldsymbol{x}) = \tfrac{1}{2} (s_t^2 - 1) \dot{\boldsymbol{\Sigma}}_t \frac{\nabla_{\boldsymbol{x}} p_t(\boldsymbol{x})}{p_t(\boldsymbol{x})}$$

$$\boldsymbol{f}_t(\boldsymbol{x}) = \tfrac{1}{2} (s_t^2 - 1) \dot{\boldsymbol{\Sigma}}_t \nabla_{\boldsymbol{x}} \log p_t(\boldsymbol{x}).$$

Substituting in the forward (A.2) and backward (A.58) SDE we get, respectively,

$$d\boldsymbol{x}_+ = \tfrac{1}{2} (s_t^2 - 1) \dot{\boldsymbol{\Sigma}}_t \nabla_{\boldsymbol{x}} \log p_t(\boldsymbol{x}) \, dt + \boldsymbol{G}_t \, d\boldsymbol{\omega}_t, \tag{A.65}$$

$$d\boldsymbol{x}_- = (s_t^2 + \tfrac{1}{2}) \dot{\boldsymbol{\Sigma}}_t \nabla_{\boldsymbol{x}} \log p_t(\boldsymbol{x}) \, dt + \boldsymbol{G}_t \, d\tilde{\boldsymbol{\omega}}_t. \tag{A.66}$$

Since, from (A.9), $s_t^2 \dot{\boldsymbol{\Sigma}}_t = \boldsymbol{G}_t \boldsymbol{G}_t^T$ is an equation involving diagonal matrices, we can safely write $\boldsymbol{G}_t = s_t \dot{\boldsymbol{\Sigma}}_t^{1/2}$. This leads to the SDE for diagonal $\boldsymbol{\Sigma}_t$, after the score function is written in terms of the denoiser and (A.50) is used.

**General case: full $\boldsymbol{\Sigma}_t$.**  Bringing back the drift term $\boldsymbol{f}_t(\boldsymbol{x}) = (\dot{s}_t / s_t) \, \boldsymbol{x} = (\frac{d}{dt} \log s_t) \, \boldsymbol{x}$ into (A.58), we have

$$d\boldsymbol{x}_t = \left[ -\tfrac{d}{dt} \log s_t \, \boldsymbol{x}_t + s_t^2 \dot{\boldsymbol{\Sigma}}_t \, \nabla_{\boldsymbol{x}} \log p_t(\boldsymbol{x}_t) \right] dt + \boldsymbol{G}_t \, d\tilde{\boldsymbol{\omega}}_t,$$

$$= \left[ -\tfrac{d}{dt} \log s_t \, \boldsymbol{x}_t + s_t^2 \dot{\boldsymbol{\Sigma}}_t \, s_t^{-1} \, \boldsymbol{\Sigma}_t^{-1} \left[ D(\boldsymbol{x}_t/s_t, \boldsymbol{\Sigma}_t) - \boldsymbol{x}_t/s_t \right] \right] dt + \boldsymbol{G}_t \, d\tilde{\boldsymbol{\omega}}_t,$$

$$= \left[ -\tfrac{d}{dt} \log s_t \, \boldsymbol{x}_t + s_t \dot{\boldsymbol{\Sigma}}_t \, \boldsymbol{\Sigma}_t^{-1} \left[ D(\boldsymbol{x}_t/s_t, \boldsymbol{\Sigma}_t) - \boldsymbol{x}_t/s_t \right] \right] dt + \boldsymbol{G}_t \, d\tilde{\boldsymbol{\omega}}_t,$$

$$= -(d \log s_t) \, \boldsymbol{x}_t + s_t (d\boldsymbol{\Sigma}_t) \, \boldsymbol{\Sigma}_t^{-1} \left[ D(\boldsymbol{x}_t/s_t, \boldsymbol{\Sigma}_t) - \boldsymbol{x}_t/s_t \right] + \boldsymbol{G}_t \, d\tilde{\boldsymbol{\omega}}_t.$$

Now, for the Itô diffusion to be well defined, we need $\boldsymbol{G}_t \boldsymbol{G}_t^T = s_t^2 \dot{\boldsymbol{\Sigma}}_t$ to be positive semi-definite. We can then still write $\boldsymbol{G}_t = s_t \dot{\boldsymbol{\Sigma}}_t^{1/2}$, and take into account that finite difference approximations of $d\boldsymbol{\Sigma}_t$ have to be projected back to the positive semi-definite cone if needed.

## B    BASELINE MEAN FORECAST

In this appendix we derive the theoretical MSE and MAE of the mean forecast baseline used in our experiments. The derivation follows the classical normality assumption for forecasting errors (see, e.g., Hyndman & Athanasopoulos (2018)).

### B.1    SETUP

Let $(y_t)_{t=1}^{T}$ be a univariate time series generated as

$$y_t = \mu + \varepsilon_t, \qquad \varepsilon_t \overset{\text{iid}}{\sim} \mathcal{N}(0, \sigma^2), \tag{A.67}$$

with unknown mean $\mu$ and variance $\sigma^2$. We observe $T$ past values $y_1, \ldots, y_T$ and consider forecasting a future value $y_{T+H}$, where $H \geq 1$.

The baseline forecast we use is the sample mean

$$\hat{y}_{T+H|T} = \bar{y}_T := \frac{1}{T} \sum_{t=1}^{T} y_t, \tag{A.68}$$

which is constant across horizons $H$.

Throughout, expectations and variances are taken with respect to the joint distribution of $(y_1, \ldots, y_T, y_{T+H})$ under the model (A.67). We first derive the distribution of the forecast error $e_{T+H}$, and then obtain closed-form expressions for MSE and MAE.

### B.2    DISTRIBUTION OF THE FORECAST ERROR

The forecast error at horizon $H$ is

$$e_{T+H} := y_{T+H} - \hat{y}_{T+H|T} = y_{T+H} - \bar{y}_T. \tag{A.69}$$

Using (A.67) and (A.68), we can write

$$y_{T+H} = \mu + \varepsilon_{T+H}, \tag{A.70}$$
$$\bar{y}_T = \mu + \bar{\varepsilon}_T, \tag{A.71}$$

where

$$\bar{\varepsilon}_T := \frac{1}{T} \sum_{t=1}^{T} \varepsilon_t \sim \mathcal{N}\left(0, \frac{\sigma^2}{T}\right). \tag{A.72}$$

By independence of the innovations,

$$\varepsilon_{T+H} \sim \mathcal{N}(0, \sigma^2), \qquad \varepsilon_{T+H} \perp \bar{\varepsilon}_T. \tag{A.73}$$

Hence,

$$e_{T+H} = \varepsilon_{T+H} - \bar{\varepsilon}_T. \tag{A.74}$$

Since $e_{T+H}$ is a linear combination of independent Gaussian random variables,

$$e_{T+H} \sim \mathcal{N}\left(0, \sigma^2 \left(1 + \frac{1}{T}\right)\right). \tag{A.75}$$

Note that this distribution does not depend on the horizon $H$.

### B.3    MEAN SQUARED ERROR (MSE)

The MSE of the baseline forecast at horizon $H$ is

$$\text{MSE}_{\text{mean}}(T) := \mathbb{E}\left[e_{T+H}^2\right]. \tag{A.76}$$

Using (A.75),

$$\text{MSE}_{\text{mean}}(T) = \text{Var}(e_{T+H}) = \sigma^2 \left(1 + \frac{1}{T}\right). \tag{A.77}$$

In particular, if the time series is standardized so that $\sigma^2 = 1$, we obtain

$$\text{MSE}_{\text{mean}}(T) = 1 + \frac{1}{T}. \tag{A.78}$$

### B.4 MEAN ABSOLUTE ERROR (MAE)

The MAE of the baseline forecast at horizon $H$ is

$$\text{MAE}_{\text{mean}}(T) := \mathbb{E}\big[|e_{T+H}|\big]. \tag{A.79}$$

From (A.75), we have

$$e_{T+H} \sim \mathcal{N}(0, \tau^2), \qquad \tau^2 := \sigma^2\left(1 + \frac{1}{T}\right). \tag{A.80}$$

Let $Z \sim \mathcal{N}(0,1)$ and write $e_{T+H} = \tau Z$. Then

$$\mathbb{E}\big[|e_{T+H}|\big] = \tau \, \mathbb{E}\big[|Z|\big]. \tag{A.81}$$

It is a standard result that for a standard normal random variable,

$$\mathbb{E}\big[|Z|\big] = \sqrt{\frac{2}{\pi}}. \tag{A.82}$$

Therefore,

$$\text{MAE}_{\text{mean}}(T) = \sqrt{\frac{2}{\pi}}\, \tau = \sqrt{\frac{2}{\pi}}\, \sigma\sqrt{1 + \frac{1}{T}}. \tag{A.83}$$

In the standardized case $\sigma^2 = 1$, this simplifies to

$$\text{MAE}_{\text{mean}}(T) = \sqrt{\frac{2}{\pi}}\, \sqrt{1 + \frac{1}{T}}. \tag{A.84}$$

### B.5 SUMMARY

Under the Gaussian error model (A.67) and the mean baseline forecast (A.68), the theoretical error measures—for standardized datasets with unit variance ($\sigma^2 = 1$)—are

$$\text{MSE}_{\text{mean}}(T) = 1 + \frac{1}{T}, \tag{A.85}$$

$$\text{MAE}_{\text{mean}}(T) = \sqrt{\frac{2}{\pi}}\, \sqrt{1 + \frac{1}{T}}. \tag{A.86}$$

These expressions are used to compute the Baseline column in Table 6, where $T = H$.

## C EXPERIMENTS

### C.1 DATASETS

The ETT, Exchange, Weather, and Solar datasets are available from `https://github.com/thuml/iTransformer`, and the Stock dataset from `https://github.com/Y-debug-sys/Diffusion-TS`.

### C.2 HYPERPARAMETERS

For each dataset we tune a small set of hyperparameters on the validation split and then keep the selected configuration fixed for all reported test results. Concretely, we vary (a) the context backward shift $k_{\text{ctx}}$ used for conditional denoising (i.e. the context window is shifted to the past by $k_{\text{ctx}}$ to act like a conditioning window), (b) the clamping range $(s_{\min}, s_{\max})$ of the scale schedule $s_t$, (c) the clamping range $(v_{\min}, v_{\max})$ of the variance schedule $\Sigma_t$, and (d) the choice of denoising network architecture. The best intervals are tuned via small discrete grids on the validation set, and chosen to minimize validation MSE. We also compare two noise schedule variants—cumulative vs. sliding $\Sigma_t$—and, for each dataset, and report results using the better-performing variant. The final per-dataset hyperparameters used in all experiments are summarized in Table 7.

### C.3 DENOISER NETWORK ARCHITECTURES

We evaluate several denoising backbones of varying complexity.

Table 7: TEDM hyperparameters selected per dataset: context backward shift $k_{\text{ctx}}$, variance clamping interval $[v_{\min}, v_{\max}]$, and scale clamping interval $[s_{\min}, s_{\max}]$.

| **Dataset** | $k_{\text{ctx}}$ | $v_{\min}$ | $v_{\max}$ | $s_{\min}$ | $s_{\max}$ |
|---|---|---|---|---|---|
| ETTh1 | 0 | $5.9 \times 10^{-6}$ | 1.91 | 1.25 | 9.38 |
| ETTh2 | 1 | $9.8 \times 10^{-6}$ | 6.38 | 0.47 | 1.74 |
| ETTm1 | 4 | $5.7 \times 10^{-7}$ | 2.33 | 1.24 | 2.68 |
| ETTm2 | 0 | $7.6 \times 10^{-3}$ | 7.17 | 0.74 | 5.14 |
| Exchange | 9 | $6.9 \times 10^{-7}$ | 7.72 | 0.87 | 4.58 |
| Stock | 1 | $4.6 \times 10^{-5}$ | 7.69 | 0.11 | 2.17 |
| Weather | 1 | $4.6 \times 10^{-5}$ | 7.69 | 0.11 | 2.17 |

**LinearNet.** LinearNet is a simple fully connected layer that applies a linear transformation Linear(seq_len, seq_len) along the temporal dimension of the noised input. It does not incorporate any temporal inductive bias (e.g., recurrence or attention) and serves as a minimalist baseline to assess the impact of architectural complexity.

**UNet.** Our UNet adapts the ADM architecture from "Diffusion Models Beat GANs on Image Synthesis" (Dhariwal & Nichol, 2021) to sequential data, leveraging alias-free resampling (Anjum, 2024) and rotary embeddings (Su et al., 2023) for time series. Noise levels are embedded (via learned or sinusoidal embeddings) and mapped into a high-dimensional space through two linear layers. The encoder stacks residual 1D convolutional blocks with downsampling between resolutions and applies self-attention at selected scales. Unless otherwise specified, the UNet uses feature size $d = $ feat_size equal to the number of dataset features, Kaiser kernel size 64, and Kaiser $\beta = 14.77$, which are kept fixed across all datasets.

**ConvLSTMNet.** ConvLSTMNet combines convolutional filtering with a bidirectional LSTM to capture both local and long-range temporal dependencies. Diffusion noise is embedded via positional or learned sinusoidal mappings, and the noised signal is adapted through shift-and-scale convolutions. A lightweight pre-LSTM 1D convolution refines these features, which are then processed by a bidirectional LSTM layer.

**AttnNet.** AttnNet employs a single cross-attention layer (Vaswani et al., 2023) to enable the denoiser to leverage mutual information between the noised sequence and the conditioning context at each time step. Concretely, it uses a single multi-head attention block

$$\text{nn.MultiheadAttention(embed\_dim} = d, \text{ num\_heads} = d),$$

with the same configuration shared across all datasets.

**AttnNetSigma.** AttnNetSigma extends AttnNet by stacking two cross-attention modules: one attends from the noised input to its context, and the other attends to the noise level. Each attention block is followed by a residual connection and LayerNorm (He et al., 2015).

**AttnPosEmbNet.** AttnPosEmbNet augments cross-attention with learned time-step embeddings and Feature-wise Linear Modulation (FiLM) conditioning on the noise level (Perez et al., 2017). This design allows the denoiser to modulate its representations explicitly as a function of diffusion time.

## C.4 TRAINING

We train all models with a batch size of 128 and select hyperparameters via validation tuning separately for each dataset. Optimization is performed with Adam (Kingma & Ba, 2017), using a linear learning-rate warmup over the first 15% of epochs, followed by a reduce-on-plateau schedule. Models are trained without early stopping, and we report results from the final checkpoint evaluated on the held-out test set. All experiments are run on a single machine equipped with 8 NVIDIA Tesla A100 GPUs (40 GiB each). To facilitate exact reproducibility, we fix random seeds for data shuffling and parameter initialization.

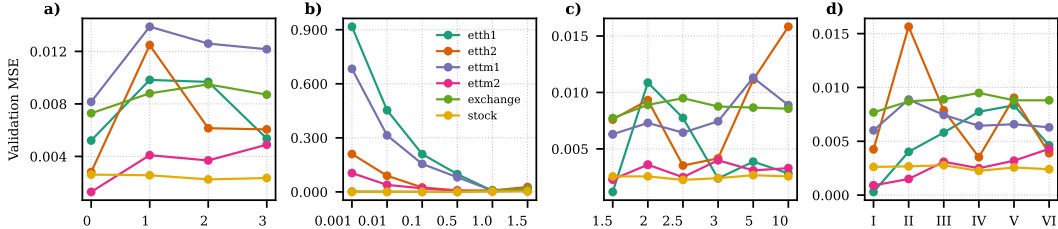

Figure 5: Ablation study on min-max normalized validation MSE across different experimental configurations and datasets. Subplots (a)–(d) correspond to: (a) context backward shift for conditionally denoising, (b) minimum value for clamping $s_t$, (c) maximum value for clamping $s_t$, and (d) different network architectures (see SM): Roman numerals I–VI denote, in the respective order, the following architectures: AttnNet, AttnSigmaNet, AttnPosEmbNet, ConvLSTMNet, LinearNet and UNet.

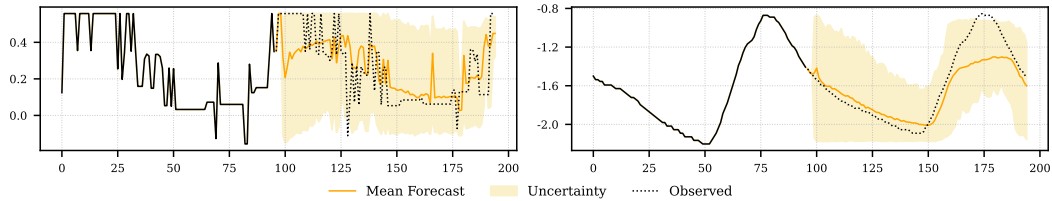

Figure 6: Probabilistic forecasts on the ETTm2 dataset, using the SDE of section A.5.2. The plots show the predicted mean, uncertainty bounds (95% prediction intervals), and ground-truth values for two representative time series.

## C.5 OTHER ABLATION STUDIES

To identify the most influential hyperparameters within our diffusion framework, we conducted systematic ablations over various parameters and architectures. In our ablation study, for each parameter, we measured validation MSE across a grid of candidate values. Parameters exhibiting the strongest correlation with forecasting accuracy were selected for further processing. Using such parameters, we did fine-grained tuning to obtain our best results in Table 2. These results were obtained with the UNet, with the lightweight architectures (e.g. AttnNet or LinearNet) still delivering SOTA performance with minimal compute—in the datasets in which we outperform.

We designed multiple ablation studies to get more insight about TEDM. The most significant studies are shown in Fig. 5. For conditionally denoising, we use a conditioning window obtained from the given window by striding backwards by a predefined number of steps $k_{\text{ctx}}$. We notice in Fig. 5(a) that this may hinder performance.

We also studied clamping of values in the scaled schedule $s_t$. Since we compute it from $\mathbb{E}(\boldsymbol{x}_t) = s_t \boldsymbol{x}_0$, in the cumulative evaluation—element-division of the starting point from the cumulative average—the division can blow up for data close to zero. Figs. 5 (b) & (c) show that there is more sensitivity to the minimum values than to the maximum used for clamping.

Finally, we considered several denoiser architectures of varying space complexity (discussed in the SM). Most remarkably, using just a Linear layer with space complexity $O(T\,d)$ gives results (V in Fig. 5), in several datasets, comparable to the best network using self-attention between the given and context window (I in Fig. 5).

## D PROBABILISTIC FORECASTS

Karras et al. (2022) derived their SDE for stochastic sampling from the isotropic heat equation. Our analogous SDE derivation, in section A.5.2 (for the anisotropic case), theoretically relies on the assumption of diagonal $\boldsymbol{\Sigma}_t$. Examples of the skill when sampling from that SDE is shown quantitatively in Table 8 and qualitatively in Fig. 6. As seen, TEDM's probabilistic calibration (CRPS/QICE) lags behind most of the other methods.

Table 8: CRPS and QICE for probabilistic forecasts with the SDE in section A.5.2 (prediction horizon $H = 96$). Datasets: ETTh2, Exchange. Lower is better.

| Methods | Metric | ETTh2 | Exchange |
|---|---|---|---|
| TimeDiff | CRPS | 0.380 | 0.287 |
| | QICE | 0.142 | 0.099 |
| DiffusionTS | CRPS | 1.122 | 1.232 |
| | QICE | 0.095 | 0.087 |
| TMDM | CRPS | 0.393 | 0.258 |
| | QICE | 0.038 | 0.049 |
| NsDiff | CRPS | **0.349** | **0.222** |
| | QICE | **0.025** | **0.038** |
| **TEDM** | CRPS | 0.589 | 0.775 |
| | QICE | 0.093 | 0.111 |

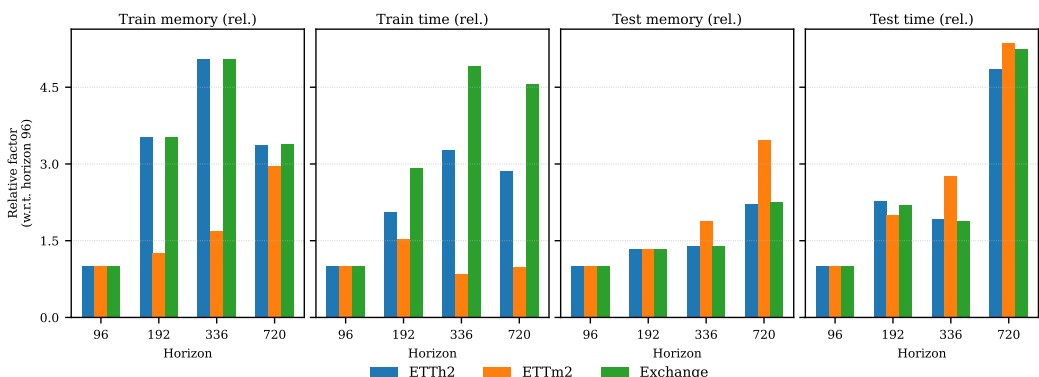

Figure 7: Relative per-batch training and inference memory (MB) and time (s) for TEDM across datasets and forecast horizons, normalized by the cost at horizon $H = 96$. All measurements are obtained on the same hardware and with a fixed batch size. Apparent drops in cost at larger horizons compared to the preceding horizon are due to dropping incomplete batches, which slightly changes the number of processed batches and thus the reported averages.

The fact that deterministic sampling outperforms other methods in point-forecast skill, while probabilistic sampling underperforms is intriguing. Our hypothesis is that the inference rule for deterministic sampling (derived in section A.5.1) is more general and hence the diagonal approximation of $\Sigma_t$ better represents cases with weakly correlated features. To test this hypothesis, we introduce a novel method to sample quantiles of the predictive distribution by only using TEDM's *deterministic* inference rule. Preliminary results of this method (explained in detail in a future publication) are shown in Table 9. It shows promising results in probabilistic forecast, being competitive with NsDiff.

Table 9: Probabilistic skill by sampling quantiles using ODE (prediction horizon $H = 96$). Datasets: ETTh2, Exchange. Lower is better.

| Methods | Metric | ETTh2 | Exchange |
|---|---|---|---|
| NsDiff | CRPS | 0.349 | 0.222 |
| | QICE | **0.025** | **0.038** |
| **TEDM** | CRPS | **0.294** | **0.186** |
| | QICE | 0.040 | 0.093 |

## E LONGER HORIZONS

We reproduce the primary diffusion baselines table for a subset of datasets (ETTh2, ETTm2, Exchange) and update TEDM with the best of the two noise schedule variant (cumulative/sliding $\Sigma_t$) provided; other methods follow the same evaluation protocol as in the main text.

Table 10: MSE and MAE scores for diffusion-based forecasting methods with horizon $H = 192$. TEDM uses the best of the noise schedule variant (cumulative/sliding $\Sigma_t$) per dataset. Lower is better.

| Methods | Metric | ETTh2 | ETTm2 | Exchange |
|---------|--------|-------|-------|----------|
| TimeDiff | MSE | 0.364 | 0.209 | 0.208 |
|          | MAE | 0.393 | 0.296 | 0.331 |
| DiffusionTS | MSE | 3.017 | 3.517 | 3.302 |
|             | MAE | 1.340 | 1.472 | 1.493 |
| TMDM | MSE | 0.564 | 0.313 | 0.212 |
|      | MAE | 0.517 | 0.350 | 0.338 |
| ARMD | MSE | 0.311 | 0.181 | **0.093** |
|      | MAE | **0.338** | **0.255** | **0.203** |
| NsDiff | MSE | 0.460 | 0.250 | 0.146 |
|        | MAE | 0.452 | 0.328 | 0.280 |
| **TEDM** | MSE | **0.260** | **0.163** | 0.153 |
|          | MAE | 0.354 | 0.282 | 0.276 |

Table 11: TEDM robustness over 4 random seeds at horizon 96. Reported are mean $\pm$ std over seeds.

| Dataset | MSE | MAE |
|---------|-----|-----|
| ETTh1 | $0.598 \pm 0.002$ | $0.526 \pm 0.001$ |
| ETTh2 | $0.216 \pm 0.001$ | $0.320 \pm 0.001$ |
| ETTm1 | $0.419 \pm 0.003$ | $0.442 \pm 0.002$ |
| ETTm2 | $0.137 \pm 0.001$ | $0.254 \pm 0.000$ |
| Exchange | $0.069 \pm 0.000$ | $0.184 \pm 0.001$ |
| Solar | $1.108 \pm 0.034$ | $0.721 \pm 0.042$ |
| Stock | $0.055 \pm 0.001$ | $0.180 \pm 0.002$ |
| Weather | $0.225 \pm 0.005$ | $0.268 \pm 0.008$ |

On ETTh2 and ETTm2, TEDM achieves the best MSE among the compared diffusion methods, improving over ARMD while also delivering strong MAE (second-best behind ARMD). On Exchange, ARMD remains the most accurate on both MSE and MAE, with NsDiff second on MSE and TEDM a close third. Overall, these results indicate that TEDM remains competitive for longer forecast horizons.

To characterize computational scaling over longer horizons, Fig. 7 reports relative per-batch training and inference time/memory for TEDM as a function of forecast horizon (normalized to the cost at horizon 96). We observe only moderate growth with horizon on different dataset. This indicates that TEDM remains practical for long-horizon forecasting.

## F ROBUSTNESS

All TEDM results are averaged over 4 random seeds (different data shuffles and parameter initializations); we report mean values in the main tables, and mean ± standard deviation in Table 11. Our method shows low variance across seeds, indicating stable training and inference.

## G MORE FORECAST SAMPLES AND FAILURE CASES

Figure 8 shows TEDM forecasts on eight benchmark datasets. Each row corresponds to a dataset and each column to a randomly selected test window and feature. Across smoother series (ETTh1, ETTh2, ETTm2), TEDM tracks level, trend, and seasonality, while on more volatile datasets (Exchange,

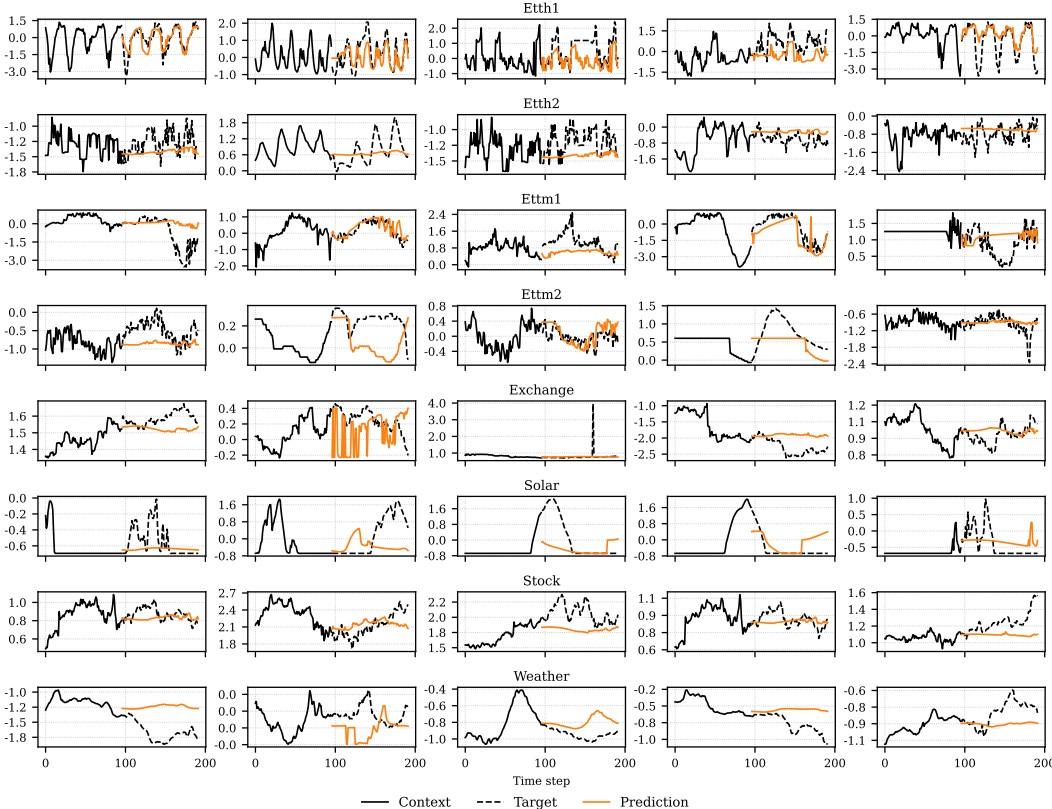

Figure 8: Qualitative TEDM forecasts across eight benchmark datasets. Each panel shows a randomly sampled test window and feature: black solid lines are input histories, black dashed lines are ground-truth futures, and orange lines are TEDM forecasts. Time is shown as input followed by forecast steps.

Solar-Energy, Stock) it still captures the overall direction and scale of movements. These examples qualitatively support the quantitative gains reported in our main results.

