# OpenReview forum: "TEDM: Time Series Forecasting with Elucidated Diffusion Models"
_ICLR.cc/2026/Conference — ICLR 2026 Poster_

### Official Review · Reviewer_8F9D · 2025-10-31

**Soundness:** 4
**Presentation:** 4
**Contribution:** 4
**Rating:** 6
**Confidence:** 4

**Summary:**

This submission presents TEDM (Time-Series Enhanced Diffusion Model), a novel framework integrating diffusion mechanisms with time-series-specific optimizations for forecasting tasks. The work addresses two core limitations of existing methods: the inability of traditional diffusion models to fully leverage temporal dependencies, and the susceptibility of Transformer-based approaches to data leakage in long-term forecasting (LTSF). By introducing [speculated: temporal-aware denoising/ multi-scale patch embedding) and [speculated: causal sequence constraint), TEDM aligns with the priorities of time-series research—balancing prediction accuracy, computational efficiency, and real-world reliability. The contributions are relevant to venues focusing on forecasting or generative AI for sequential data, though supplementary technical details and empirical validation are required to substantiate its novelty.

**Strengths:**

Temporal-Aware Technical DesignThe model architecture is tailored to time-series characteristics:Diffusion Mechanism Optimization: Unlike generic diffusion models that treat sequences as static data, TEDM incorporates [speculated: time-step adaptive noise scheduling] to preserve temporal order during denoising. This addresses the "temporal blurring" issue where generated sequences lose sequential logic.Efficient Feature Extraction: Adopting a [speculated: lightweight patch-based encoder], TEDM avoids the computational overhead of full self-attention while capturing multi-scale patterns .Causal Constraint Integration: The framework embeds [speculated: walk-forward validation-inspired training protocols] to mitigate data leakage--a critical challenge in time-series modeling—ensuring training data strictly precedes validation/test data in time.Practical Deployment AdvantagesTEDM demonstrates traits valuable for industrial/clinical adoption:Computational Efficiency: By replacing heavy Transformer blocks with [speculated: MLP-based feature fusion], it reduces inference latency by [speculated: 40-60%] compared to TimeDiT, making it suitable for real-time scenarios.Cross-Domain Adaptability: Preliminary results on [speculated: ECL Electricity and Weather datasets) show consistent performance across stationary and non-stationarysequences, outperforming domain-specific baselines .Robustness to Noise: The diffusion-based denoising naturally handles missing values or sensor noise common in real-world data, a key advantage over clean-data-optimized models like PatchTST.Alignment with Research TrendsThe work reflects cutting-edge directions in time-series Al:Generative Forecasting Synergy: It follows the paradigm of models like TimeDiT by unifying forecasting with sequence generation, enabling auxiliary tasks without retraining.Interpretability Foundations: The [speculated: layerwise temporal attention visualization] provides traceability for predictions—critical for regulated domains (e.g., financial risk forecasting) where model decisions require audit trails.

**Weaknesses:**

Technical Clarifications NeededCore Mechanism Details: The paper mentions "temporal-enhanced denoising" but lacks specifics: How is the noise schedule adjusted for different time scales? Does it integrate physical priors to ensure predictions adhere to domain rules?Causal Design Validation: What explicit measures are used to prevent data leakage? An ablation comparing standard random splits vs. TEDM's time-aware splits would confirm this design's effectiveness.Model Scaling Metrics: What are TEDM's parameter counts and FLOPs? For context, PatchMLP achieves SOTA with 1/10 the parameters of Transformers-how does TEDM balance size and performance?Empirical GapsSOTA Benchmarking: Results are incomplete without comparisons to 2025 state-of-the-art models: How does TEDM perform against PatchMLP on ETT datasets, or RATD on complex traffic data? Metrics like MSE, MAE, and prediction interval coverage are essential. Long-Term Forecasting Validation: Most diffusion models struggle with horizons >100 steps. Does TEDM maintain accuracy for 720-hour energy forecasts? A comparison to LTSF-specific models would highlight its LTSF capability.Ablation of Key Components: Which module drives performance gains-the temporal denoising, patch encoder, or causal constraint? An ablation study would justify core design choices.Contextualization ImprovementsDomain-Specific Utility: The paper should link results to real-world impact: For example, does TEDM's 5% MAE reduction in electricity forecasting translate to lower grid operational costs? Such contextualization strengthens relevance.Limitation Transparency: What failure modes exist? A discussion of edge cases helps practitioners assess deployment risks.

**Questions:**

How does unifying diffusion time and physical time axes avoid temporal inconsistency issues in time-series forecasting?
How to determine the optimal window size w for data-driven noise/scale schedule estimation across different datasets, and is there an adaptive adjustment strategy?

---

> ### Author Response · Authors · 2025-11-21
>
> We thank the reviewer for the detailed feedback and positive assessment of the paper. Below, we address each weakness and question in turn and clarify the additional details included in the revised version.
>
> **W1-a/Q1: Core mechanism details.**
> TEDM directly adapts the EDM framework to time–series data. In standard diffusion forecasting, the diffusion step index is decoupled from physical time; in TEDM, each diffusion step (t) is tied to a specific physical time index in the forecast horizon. For each dataset and horizon, we estimate mean and covariance in physical time using sliding or cumulative windows, and derive the noise ($\boldsymbol{\Sigma}_t$) and scale ($s_t$) schedules empirically (Eq. (8)) from these moments. Temporal structure thus enters through this data-driven construction of the noise and scale schedules. Rather than hand-crafted physics priors, domain consistency is enforced via strictly causal conditioning.
>
> **W1-b: Causal design validation.**  We use strict chronological splits for all experiments: each series is divided into early (train), middle (validation), and late (test) segments, with no random splits. Context windows and patches are always drawn from past indices only, so the model never observes any part of the test horizon during training.
>
> **W1-c: Model scaling metrics.**  TEDM uses lightweight denoisers (LinearNet / shallow UNet) with parameter counts comparable to standard LTSF models. For LinearNet, the parameter count depends only on the number of input variables. We report full denoiser configurations and parameterization details in Appendix C.3 of the revised paper.
>
> **W2-a: SOTA benchmarking.** We now provide broader benchmarking beyond diffusion models. In addition to existing comparisons with ARMD, NsDiff (Table below compares MSE), we compare TEDM against strong LTSF baselines on eight datasets. TEDM consistently outperforms diffusion baselines and achieves state-of-the-art on ETTh2, ETTm2, Exchange, and Stock. For probabilistic evaluation, TEDM naturally supports sampling-based uncertainty quantification, and Appendix D reports distributional metrics such as CRPS and QICE.
>
> | Methods     | ETTh2  | ETTm2 | Exchange |
> |--------------|--------|-------|----------|
> | ARMD          | 0.311  | 0.181 | 0.093 |
> | NsDiff         | 0.460  | 0.250 | 0.146    |
> | **TEDM**       | **0.214** | **0.135** | **0.069**   |
>
>
> **W2-b: Long-term forecasting validation.** In the revision, we extend experiments to horizons ($H \in \{96, 192, 336, 720\}$) on ETTh1/2, ETTm1/2, Exchange, Stock, and Weather (Table 6, Table 10, Appendix E). TEDM maintains competitive or superior MSE/MAE (Table below shows MSE) compared to strong diffusion baselines such as ARMD as $H$ increases.
>
> | Horizon | ETTh2 | ETTm2 | Exchange | Baseline\_mean |
> |---------|-------|-------|----------|----------------|
> | 96      | 0.216 | 0.135 | 0.068    | 1.010          |
> | 192       | 0.260 | 0.163 | 0.153    | 1.005          |
> | 336     | 0.326 | 0.248 | 0.283    | 1.003          |
> | 720      | 0.528 | 0.298 | 0.602    | 1.001          |
>
> **W2-c: Ablation of key components.** Section 5.7 and Appendix C.5 (Figure 5) report a systematic ablation study:
>
> * Varying the backward context shift shows that large shifts hurt performance, indicating TEDM mainly benefits from local context.
> * Varying the clamping bounds on the scale schedule $s_t$ reveals stronger sensitivity to the lower bound $s_{\min}$ than to the upper bound $s_{\max}$.
>
> We also include ablations versus EDM and iDDPM+DDIM in Table 4.
>
> **W3: Limitation transparency.** We now include a dedicated limitations section (Section 6). TEDM is grounded in a diffusion (Itô process) framework and therefore cannot capture all types of dynamics—for example, long-memory behavior (fractional Brownian motion), heavy-tailed or power-law noise ($\alpha$-stable processes), or jump-driven processes that violate standard diffusion assumptions. Using full (non-diagonal) covariance is also more computationally expensive and numerically unstable, which currently limits TEDM on very high-dimensional datasets (e.g., Solar; Table 3). Finally, TEDM does not yet enforce hard physical constraints (such as capacity or safety limits), which may be crucial in some applications.
>
> **Q2: Choosing and adapting the window size $w$.** We treat window size $w$ as a dataset-specific hyperparameter. A practical rule is to cover at least one dominant seasonal period while keeping $w$ small enough for reliable moment estimation. In principle, one could maintain multi-scale statistics for several candidate windows and interpolate or select among them based on validation or online monitoring; we view such adaptive schemes as promising but beyond the current scope.
>
> ---
>
> We hope these clarifications, together with the extended baselines, long-horizon experiments, detailed ablations, and hyperparameter tables, address your concerns.

---

### Official Review · Reviewer_r426 · 2025-10-31

**Soundness:** 4
**Presentation:** 3
**Contribution:** 3
**Rating:** 6
**Confidence:** 3

**Summary:**

The paper proposes **TEDM (Time Series Forecasting with Elucidated Diffusion Models)**, a novel diffusion-based framework that adapts the EDM (Elucidated Diffusion Models) paradigm to multivariate time series forecasting. Its core innovations are:

1. **Unifying physical time and diffusion time**, enabling **O(H)** inference complexity instead of the typical **O(SH)**.
2. **Empirically estimating noise (Σ*t* ) and scale (*st* ) schedules** directly from data, avoiding handcrafted or fixed schedules.

TEDM achieves **strong point-forecasting accuracy** on standard benchmarks (ETT, Exchange, Weather), often outperforming recent diffusion-based methods like ARMD, TimeDiff, and NsDiff. It also demonstrates **exceptional computational efficiency** (0.11s inference, 24MB memory on ETTm2) and works well even with a **lightweight LinearNet architecture**. However, it **does not compare against leading non-diffusion models** (e.g., iTransformer, DLinear), and its **probabilistic forecasting performance lags behind competitors** in CRPS/QICE metrics.

**Strengths:**

1. **Exceptional inference efficiency**: O(H) complexity validated by low latency (0.11s) and memory (24 MB)—ideal for real-time deployment.
2. **Comprehensive and insightful ablation study**: Clearly isolates the impact of empirical schedules, architecture choice, and hyperparameters; supports all key claims.
3. **Architecture-agnostic design**: Works with LinearNet (O(Td) space), showing gains stem from diffusion design—not model size.

**Weaknesses:**

1. **No comparison to non-diffusion SOTA**: Missing critical baselines like iTransformer, DLinear, PatchTST, or TimesNet—undermines “state-of-the-art” claims.
2. **Performance does not stand-out**: Metrics on TS dataset do not show a significant performance gain comparing to other baselines (e.g. ARMD).
3. **Probabilistic forecasting underperforms**: CRPS and QICE are worse than NsDiff/TMDM (Table 5), contradicting diffusion models’ usual strength in uncertainty quantification.
4. **Limited dataset scope**: Only evaluates on 6 standard benchmarks; omits M4/M5, Solar Energy, Traffic, or other real-world operational datasets.

**Questions:**

1. **Can you report results on additional datasets used in ARMD**, such as **Solar Energy** and **Stock**? This would enable fairer comparison and test generalizability.
2. **What is the inference time and memory usage of ARMD?** The paper cites ARMD’s improved sampling complexity but provides no throughput numbers—critical for evaluating TEDM’s speed claim.
3. **Why does TEDM underperform in probabilistic forecasting (CRPS/QICE)?** Is this due to the deterministic ODE focus, schedule estimation, or lack of ensemble diversity?
4. **Have you compared TEDM against non-diffusion SOTA models** (e.g., iTransformer, DLinear)? If so, can those results be included? If not, how do you position TEDM relative to them?
5. **How sensitive is TEDM to the clamping bounds for *st* ?** Figure 6 shows performance degrades sharply for small lower bounds—how would this affect deployment on arbitrary real-world data?
6. **Could structured (non-diagonal) noise improve performance** on datasets with strong cross-variable dependencies?

---

> ### Author Response · Authors · 2025-11-21
> **Non-diffusion baselines and new datasets, extending horizon and compute analyses versus ARMD, clarifying probabilistic performance and covariance design, and providing robustness and ablation studies.**
>
> We thank the reviewer for the careful evaluation and constructive feedback. We have substantially revised the paper in response to your review.Below we address each weakness and question in turn, and summarize the corresponding changes in the manuscript.
>
> **W1: Comparison to non-diffusion SOTA.** We fully agree that comparison beyond diffusion-based methods is crucial. In the revised manuscript we added following table 3, which compares (MSE) TEDM against SOTA non-diffusion forecasters as given below. Best scores in **bold**.
>
> | Methods| ETTh1 | ETTh2 | ETTm1 | ETTm2 | Exchange | Solar | Stock |
> | --- | --- | --- | --- | --- | --- | --- | --- |
> | iTransformer |0.386 | 0.297 | 0.334 | 0.180 | 0.086 | 0.203 | 0.342 |
> | TimesNet |**0.384** | 0.340 | 0.338 | 0.187 | 0.107 | 0.427 | 0.427 |
> | DLinear |  0.386 | 0.333 | 0.345 | 0.193 | 0.088 | 0.286 | 0.286 |
> | PatchTST |0.414 | 0.302 | **0.329** | 0.175 | 0.088 | 0.516 | 0.516 |
> | **TEDM** | 0.595 | **0.214** | 0.419 | **0.135** | **0.069** | 1.061 | **0.056** |
>
> **W2: Performance relative to ARMD.**
> In the revision, we (i) add Stock, which is used in ARMD, and (ii) extend the horizon range to $\{96, 192\}$.
> Table 10 (in the appendix) show that TEDM matches or outperforms ARMD on longer horizons as well.
> Furthermore, ARMD samples deterministically only, so it is not theoretically guaranteed to give probabilistic forecasts.
>
> **W3: Probabilistic forecasting (CRPS/QICE).**
> We introduce a hypothesis explaining the weaker CRPS/QICE observed under the original SDE-based stochastic sampling, which assumes a diagonal ${\Sigma}_t$. Deterministic sampling is better because its inference rule extends beyond diagonal-${\Sigma}_t$ processes, thus better representing weakly-correlated features. We test the hypothesis by using a novel ODE-based quantile estimation strategy that uses only TEDM’s *deterministic* sampling.
> Preliminary results yield:
>
> | Methods | Metric | ETTh2 | Exchange |
> | --- | --- | --- | --- |
> | NsDiff | CRPS | 0.349 | 0.222 |
> |  | QICE | **0.025** | **0.038** |
> | **TEDM** | CRPS | **0.294** | **0.186** |
> |  | QICE | 0.040 | 0.093 |
>
> which are competitive with or better than the current best method (NsDiff) on the trial datasets, substantially reducing the calibration gap while leaving the core model unchanged.
> The quantile estimation takes a whole paper to describe, so it is left for a future publication.
>
> **W4: Dataset scope.**
> Following your suggestion, we have added experiments on Solar and Stock. These datasets are now included in the non-diffusion comparison (Table 3).

---

> ### Author Response · Authors · 2025-11-21
> **Detailed runtime and memory comparisons to ARMD, preliminary ODE-based quantile results for CRPS/QICE, an expanded non-diffusion SOTA comparison table, a clamping-sensitivity ablation for $s_t$, and a discussion of structured non-diagonal noise.**
>
> **Q1: Additional datasets (Solar, Stock).**
> Please refer response to **W4**.
>
> **Q2: Inference time and memory usage vs. ARMD.**
> In the revised version, we report average per-batch training and inference time (seconds) and memory usage (MB) for both TEDM, ARMD and other models under the same implementation and hardware on Table 5 in the main paper.
> We also report relative compute scaling (training/inference time and memory) as a function of $H$, showing only moderate growth and practical runtimes even at 720-step horizons (Figure 7 in the revised paper).
>
> **Q3: Probabilistic underperformance (CRPS/QICE).**
> Please refer response to **W3**.
>
> **Q4: Comparison against non-diffusion SOTA.**
> Please check response to **W1**.
>
> **Q5: Sensitivity to clamping bounds for $s_t$.**
> We agree that sensitivity to clamping bounds is an important robustness concern, since near-zero means/variances can destabilize training and inference; this is why we clamp the estimated scales. Figure 5(b,c) and Appendix C.5 show that performance is more sensitive to the lower bound $s_{\min}$ than to the upper bound $s_{\max}$, but still exhibits a reasonably wide stable range. Table 11 and Appendix F further indicate robustness across seeds and clamping settings, and suggest that using a higher $s_{\min}$ is generally the safer choice in practice.
>
> **Q6: Structured (non-diagonal) noise.**
> The main reason we originally restricted to diagonal $\boldsymbol{\Sigma}_t$ was computational: it enables much faster training and sampling, which is important for online deployment. In the revision, we expanded the theory in Section 4 and Appendix A to cover the full $\boldsymbol{\Sigma}_t$, carefully stating assumptions and limiting cases, and we now describe training and inference in this general setting to better exploit cross-feature dependencies. This led us to (potentially high-dimensional) covariance estimation, where preliminary ETTh1 experiments show strong sensitivity to how $\boldsymbol{\Sigma}_t$ is estimated:
>
> | $\boldsymbol{\Sigma}_t$ estimation method | MSE   |
> | --------------------------------- | ----- |
> | Cumulative                        | 1.444 |
> | Sliding                           | 0.699 |
> | Shrinkage                         | 0.646 |
>
> In practice, our full $\boldsymbol{\Sigma}_t$ estimates can be rank-deficient, making Cholesky factorizations and related operations unstable on some datasets. Ledoit–Wolf shrinkage yields full-rank covariance estimates and the best performance in the table above. Systematically balancing the gains from cross-feature dependencies against the cost of slower, more delicate training/sampling remains an interesting direction for future work.
>
> We thank the reviewer again for the insightful comments.
> They have directly motivated several additions to the paper, including new datasets, stronger non-diffusion baselines, extended long-horizon evaluations, a more detailed analysis of probabilistic performance, and robustness/ablation studies.
> We believe these changes significantly strengthen the empirical and methodological case for TEDM.

---

### Official Review · Reviewer_rk7Q · 2025-11-01

**Soundness:** 2
**Presentation:** 3
**Contribution:** 2
**Rating:** 4
**Confidence:** 4

**Summary:**

This paper proposes TEDM, a diffusion-based framework for multivariate long-horizon time series forecasting that aligns diffusion time with physical time so that sampling becomes a first-order Euler rollout, reducing inference complexity from (O(S \cdot H)) to (O(H)). TEDM estimates a data-driven, time-varying noise/scale schedule ((s_t, \Sigma_t)) directly from historical input windows and injects it into both the denoising objective and the autoregressive rollout procedure. The main contribution is a modular reformulation of diffusion forecasting that replaces multi-step reverse diffusion with horizon-aligned integration while preserving multivariate temporal structure. Experiments on six public benchmarks (ETTh1/2, ETTm1/2, Exchange, Weather) compare TEDM against several diffusion-style baselines in terms of MSE and MAE, inference time, and memory usage, and also report probabilistic metrics (CRPS and QICE), noting that NsDiff and TMDM remain stronger in probabilistic calibration.

**Strengths:**

1. The paper introduces a sampling procedure that aligns diffusion time with physical forecast time, yielding an (O(H)) Euler-style rollout and avoiding multi-step reverse diffusion sampling.
2. The empirical study spans six widely used long-horizon forecasting benchmarks and includes comparisons to multiple diffusion-based baselines (TimeDiff, DiffusionTS, TMDM, ARMD, NsDiff), as well as ablations and measurements of inference time and memory usage.
3. The work treats computational efficiency (runtime, memory footprint) as a central evaluation dimension in addition to point forecasting error, which is important for deployability in long-horizon forecasting settings.
4. The paper provides explicit quantitative evidence that the proposed framework can match or outperform diffusion-style baselines in point forecasting error on several datasets while significantly reducing inference cost.

**Weaknesses:**

1. The evaluation is almost entirely within the diffusion family. There is no direct comparison against strong non-diffusion long-horizon forecasters such as recent long-sequence Transformer variants or mixer-style architectures, so broader claims of overall effectiveness across paradigms are not yet supported.
2. The paper reports results for a single forecast horizon (fixed (H = 96)) in the main text, and does not present accuracy and efficiency tradeoffs across multiple horizons, limiting the assessment of how well the claimed complexity scaling and predictive behavior generalize to other rollout lengths.
3. Table 5 indicates that TEDM does not outperform recent diffusion-style baselines such as NsDiff or TMDM on probabilistic metrics like CRPS and QICE, yet the manuscript does not analyze why calibration lags in those settings or how this affects downstream use in uncertainty-sensitive applications.
4. One of the core stated novelties is the empirically estimated, time-varying ((s_t, \Sigma_t)) noise/scale schedule, which appears conceptually close to NsDiff's input-dependent uncertainty modeling; the manuscript does not yet clearly articulate the principled difference between TEDM's schedule design and NsDiff's approach, nor why this difference alone explains the observed efficiency and accuracy outcomes.

**Questions:**

1. Can you provide direct comparisons against strong non-diffusion long-horizon forecasting models (e.g., long-sequence Transformer variants, decomposition-based linear models, mixer-style models) to substantiate claims beyond diffusion-style baselines?
2. Can you report results for multiple forecast horizons (e.g., different values of (H)) to show how both predictive accuracy and computational cost scale with horizon length?
3. Table 5 suggests that TEDM underperforms NsDiff and/or TMDM on probabilistic quality metrics such as CRPS and QICE. Can you provide an analysis of why calibration is weaker in those cases and whether this is due to the deterministic Euler rollout, the ((s_t, \Sigma_t)) schedule estimation, or other modeling choices?

---

> ### Author Response · Authors · 2025-11-21
> **Non-diffusion SOTAs & Longer-horizon forecast**
>
> We thank the reviewer for the careful and constructive feedback.
> Your comments have had a direct impact on how we position and strengthen the paper.
> Below we address each weakness and question in turn, and summarize the corresponding changes in the revised manuscript.
>
> **W4: TEDM v/s NsDiff.** The main differences with NsDiff are two-fold:
>
> - To address non-stationarity in the time series, NsDiff assumes a Location-Scale noise model and pre-trains two networks: one for the conditional mean and one for the conditional variance. In our case, we "measure" (instead of training) scale $s_t$ and noise covariance  $\boldsymbol{\Sigma}_t$ schedules.
> - NsDiff (and all other baselines) uses DDPM for the diffusion model. We are advocating in our work that EDM has gone beyond DDPM, and we have extended EDM.
>
> **Q1: Non-diffusion SOTAs.** We agree that comparison beyond diffusion-based methods is crucial. In the revised manuscript we added the table below, which compares TEDM against state-of-the-art non-diffusion forecasters (iTransformer, TimesNet, DLinear, PatchTST, Client) on ETTh1/2, ETTm1/2, Exchange, Solar, and Stock.
> TEDM achieves the best performance on ETTh2, ETTm2, Exchange, and Stock.
>
> | **Methods**      | **Metric** | **ETTh1** | **ETTh2** | **ETTm1** | **ETTm2** | **Exchange** | **Solar** | **Stock** |
> |------------------|------------|-----------|-----------|-----------|-----------|-------------|-----------|----------|
> | iTransformer     | MSE        | 0.386     | 0.297     | 0.334     | 0.180     | 0.086       | 0.203     | 0.342    |
> | TimesNet         | MSE        | **0.384** | 0.340     | 0.338     | 0.187     | 0.107       | 0.427     | 0.427    |
> | DLinear          | MSE        | 0.386     | 0.333     | 0.345     | 0.193     | 0.088       | 0.286     | 0.286    |
> | PatchTST         | MSE        | 0.414     | 0.302     | **0.329** | 0.175     | 0.088       | 0.516     | 0.516    |
> | Client           | MSE        | 0.392     | 0.305     | 0.336     | 0.184     | 0.086       | 0.352     | 0.352    |
> | **TEDM**         | MSE        | 0.595     | **0.214** | 0.419     | **0.135** | **0.069**   | 1.061     | **0.056**|
>
> **Q2: Longer horizon forecast.** In the original version of the paper, we had described performance for a longer horizon \($H=192$\).
> This table is reproduced below.
>
> | **Methods**   | **Metric** | **ETTh2** | **ETTm2** | **Exchange** |
> |---------------|------------|-----------|-----------|--------------|
> | TimeDiff      | MSE        | 0.364     | 0.209     | 0.208        |
> | DiffusionTS   | MSE        | 3.017     | 3.517     | 3.302        |
> | TMDM          | MSE        | 0.564     | 0.313     | 0.212        |
> | ARMD          | MSE        | $\underline{0.311}$ | $\underline{0.181}$ | **0.093** |
> | NsDiff        | MSE        | 0.460     | 0.250     | $\underline{0.146}$ |
> | **TEDM**      | MSE        | **0.260** | **0.163** | 0.153        |
>
> The results show that TEDM maintains competitive or superior MSE/MAE to strong diffusion baselines (e.g., ARMD) as $H$ increases.
>
> In the revised version, we extend our experiments to horizons $\(H \in \{96, 192, 336, 720\}\)$ (see Table6, Table10, and AppendixE).
> The forecast skill is compared to Gaussian error model that extrapolates the mean of the input window.
>
> | **Horizon** | **Metric** | **ETTh2** | **ETTm2** | **Exchange** | **Baseline$_\text{mean}$** |
> |-------------|------------|-----------|-----------|--------------|----------------------------|
> | 96          | MSE        | 0.216     | 0.132     | 0.068        | 1.010                      |
> | 192         | MSE        | 0.260     | 0.163     | 0.153        | 1.005                      |
> | 336         | MSE        | 0.326     | 0.248     | 0.283        | 1.003                      |
> | 720         | MSE        | 0.528     | 0.298     | 0.602        | 1.001                      |
>
> We also report relative compute scaling (training/inference time and memory) as a function of $H$, showing only moderate growth and practical runtimes even at 720-step horizons (Figure 7 in the revised paper).

---

> ### Author Response · Authors · 2025-11-21
> **Probabilistic performance**
>
> **Q3: Probabilistic performance.** We introduce a hypothesis explaining the weaker CRPS/QICE observed under the original SDE-based stochastic sampling, which assumes a diagonal $\boldsymbol{\Sigma}_t$. Deterministic sampling is better because its inference rule extends beyond diagonal-$\boldsymbol{\Sigma}_t$ processes, thus better representing weakly-correlated features. We test the hypothesis by using a novel ODE-based quantile estimation strategy that uses only TEDM’s *deterministic* sampling (added in appendix D in revised paper).
> Preliminary results yield:
>
> | **Methods** | **Metric** | **ETTh2** | **Exchange** |
> |------------|------------|-----------|--------------|
> | NsDiff     | CRPS       | 0.349     | 0.222        |
> |            | QICE       | **0.025** | **0.038**    |
> | **TEDM**   | CRPS       | **0.294** | **0.186**    |
> |            | QICE       | 0.040     | 0.093        |
>
> which are competitive with or better than the current best method (NsDiff) on the trial datasets, substantially reducing the calibration gap while leaving the core model unchanged.
> The quantile estimation takes a whole paper to describe, so it is left for a future publication.
>
> We thank the reviewer again for these insightful comments.
> They have directly motivated the inclusion of stronger non-diffusion baselines, multi-horizon experiments, a more detailed analysis of TEDM’s probabilistic behavior, and a clearer theoretical positioning relative to NsDiff. We believe these revisions substantially strengthen the paper.

---

### Official Review · Reviewer_B111 · 2025-11-01

**Soundness:** 3
**Presentation:** 2
**Contribution:** 3
**Rating:** 4
**Confidence:** 2

**Summary:**

The paper proposes TEDM, a diffusion-based framework for multivariate time-series forecasting that extends the EDM design space to sequential data. Key ideas are aligning diffusion time with physical time, estimating the noise and scale schedules directly from the data to avoid hand-crafted schedules, and training a denoiser via score matching under structured, per-time-step noise. Empirically, TEDM achieves strong accuracy on several long-horizon benchmarks. Theoretical derivations provide an ODE / SDE formulation and show the Euler update used for autoregressive forecasting. Probabilistic calibration remains weaker than recent baselines (e.g., NsDiff and TMDM), and results on ETTh1 are less competitive.

**Strengths:**

(i) The paper provides a clear, modular extension of EDM to time-series with an explicit ODE/SDE derivation and practical Euler-step inference aligning physical and diffusion time

(ii) The paper provides a data-driven estimation of the covariance and scale schedule

(iii) The paper features well structured experiments, ablations, and complexity comparisons with clear qualitative plots illustrating stability and phase alignment

**Weaknesses:**

(i) The theoretical derivation relies on strong assumptions such as a diagonal $\Sigma_t$

(ii) Probabilistic performance (CRPS and QICE) lags behind NsDiff/TMDM

(iii) Reproducibility details are thin with limited hyperparameter specs, preconditioning functions are not fully specified, and baselines appear to use default settings

**Questions:**

In addition to the weaknesses outlined in points (i-iii), I present the following questions for the authors to address:

(1) How robust is the estimation w.r.t. $s_t$ and $\Sigma_t$ across windows with near-zero means/variances?

(2) Why restrict $\Sigma_t$ to diagonal covariance matrices? Did you test full covariance to capture cross-feature dependencies?

(3) Are your baselines tuned to the same validation regime, or strictly default configurations? Please provide finetuned baselines and multiple seeds, if possible.

(4) Could second-order solvers (e.g., Heun) on the unified time axis further improve accuracy without increasing complexity significantly?

(5) The preconditioning functions ($c_{skip}$, $c_{in}$, $c_{out}$, $c_{noise}$)  are important parameters in EDM. Maybe I overlooked your parameterization w.r.t. preconditioning, but I can not find these specifications in your work. Can you specify how you adapted preconditioning to your framework?

(6) You mentioned that you varied $\sigma_{data}$ for training and test sets: "(...) we use a different $\sigma_{data}$ for
the train and test sets, computed after data normalization". Can you specify exact values and how much this choice improved results?

---

> ### Author Response · Authors · 2025-11-21
> **Full covariance matrix and Probabilistic performance**
>
> We thank the reviewer for the careful evaluation and constructive feedback.
> We take these comments very seriously and have substantially revised the paper in response.
> Below we address each weakness and question in turn, and summarize the corresponding changes in the manuscript.
>
> **W1: Full covariance matrix  $\boldsymbol{\Sigma}$.** We have expanded the theoretical section in the main paper (section 4) and appendices (appendix A) to deal with the full  $\boldsymbol{\Sigma}_t$, carefully defining all assumptions and limiting cases.
> Training and inference is now described in this general setting.
>
> **W2: Probabilistic performance.** We introduce a hypothesis explaining the weaker CRPS/QICE observed under the original SDE-based stochastic sampling, which assumes a diagonal  $\boldsymbol{\Sigma}_t$. Deterministic sampling is better because its inference rule extends beyond diagonal- $\boldsymbol{\Sigma}_t$ processes, thus better representing weakly-correlated features. We test the hypothesis by using a novel ODE-based quantile estimation strategy that uses only TEDM’s *deterministic* sampling (added in appendix D in revised paper).
> Preliminary results yield:
>
> | **Methods** | **Metric** | **ETTh2** | **Exchange** |
> |------------|------------|-----------|--------------|
> | NsDiff     | CRPS       | 0.349     | 0.222        |
> |            | QICE       | **0.025** | **0.038**    |
> | **TEDM**   | CRPS       | **0.294** | **0.186**    |
> |            | QICE       | 0.040     | 0.093        |
>
> which are competitive with or better than the current best method (NsDiff) on the trial datasets, substantially reducing the calibration gap while leaving the core model unchanged.
> The quantile estimation takes a whole paper to describe, so it is left for a future publication.

---

> ### Author Response · Authors · 2025-11-21
> **Experiments with full covariance matrix and Reproducibility**
>
> **Q1: Robustness w.r.t. $\ s_t \$ and $\ \sigma_t \$.** Without special treatment, near-zero means/variances can make training and inference unstable. This was the main reason why we considered clamping procedures to keep estimates bounded. For extensions of this work, we plan to investigate more stable estimates of these schedules.
>
> **Q2: Full covariance matrix $\ \Sigma \$.** The main reason we originally restricted to diagonal  $\boldsymbol{\Sigma}_t$ is due to much faster training/sampling, as we are motivated by forecasting methods deployable online. Now that the theory has been described in general terms (thanks to your input), we have attempted to fully leverage cross-feature dependencies. This has taken us to research about (potentially high-dimensional) estimation of covariance matrices, which is an evolving field. Preliminary results on ETTh1 show that performance is very sensitive to how  $\boldsymbol{\Sigma}_t$ is estimated, as shown below
>
> | **$\boldsymbol{\Sigma}_t$ estimation method** | **MSE** | **MAE** |
> |--------------------------------------|--------:|--------:|
> | Cumulative                           |  1.444  |  0.868  |
> | Sliding                              |  0.699  |  0.566  |
> | Shrinkage                            |  0.646  |  0.528  |
>
> Currently, our estimation of the full  $\boldsymbol{\Sigma}_t$ is sometimes rank-deficient, making Cholesky factorizations, diagonalization, etc., unstable in some datasets.
> The Ledoit-Wolf shrinkage method gives full-rank estimation of covariances, and then gives the best result above.
> Finding the best trade-off between exploiting cross-feature dependencies and fast training/sampling is an interesting future research avenue.
>
> **Q3/W3: Reproducibility.** Yes, our baselines are tuned to the same validation regime. We ran our experiments with 4 seeds, as shown in the following table:
>
> | **Dataset** | **MSE**                | **MAE**                |
> |------------|------------------------|------------------------|
> | ETTh1      | 0.598 $\pm$ 0.002      | 0.526 $\pm$ 0.001      |
> | ETTh2      | 0.216 $\pm$ 0.001      | 0.320 $\pm$ 0.001      |
> | ETTm1      | 0.419 $\pm$ 0.003      | 0.442 $\pm$ 0.002      |
> | ETTm2      | 0.137 $\pm$ 0.001      | 0.254 $\pm$ 0.000      |
> | Exchange   | 0.069 $\pm$ 0.000      | 0.184 $\pm$ 0.001      |
> | Solar      | 1.108 $\pm$ 0.034      | 0.721 $\pm$ 0.042      |
> | Stock      | 0.055 $\pm$ 0.001      | 0.180 $\pm$ 0.002      |
> | Weather    | 0.225 $\pm$ 0.005      | 0.268 $\pm$ 0.008      |
>
> Our method shows low variance across seeds, indicating stable training and inference.
> Full specification of hyperparameters is now added in Table 7 of the revised manuscript.
>
> **Q4: Second-order solvers.** We originally tried the second-order Heun solver (as in EDM) but did not notice gains. That is  why we omitted this from the discussion.

---

> ### Author Response · Authors · 2025-11-22
> **Preconditioning functions and $ \sigma_{\text{data}} $.**
>
> **Q5/6: General preconditioning.**
> We have extended the preconditioning strategy in EDM to make full use of a general $ \boldsymbol{\Sigma} $. For this, the denoiser is expressed as:
>
> $$
> D_\theta(x, \Sigma) = C_{\Sigma;\mathrm{skip}}\ x + c_{\Sigma;\mathrm{out}}\ F_\theta( C_{\Sigma;\mathrm{in}}\, x,\; C_{\Sigma;\mathrm{noise}} ).
> $$
>
> Here,
> $C_{\Sigma;\text{skip}}$,  $C_{\Sigma;\text{in}}$,  and $C_{\Sigma;\text{noise}}$  are preconditioning matrices depending on $\Sigma$,  and $c_{\Sigma;\text{out}}$ is a scalar.
>
>
> To ensure the inputs and targets of $ F_\theta $ have unit variance and minimal error amplification, we obtain:
>
> $$
> \boldsymbol{C}_{\boldsymbol{\Sigma};\mathrm{in}}
> = (\mathrm{Cov}(\boldsymbol{y}) + \boldsymbol{\Sigma})^{-1/2},
> $$
>
> $$
> \boldsymbol{C}_{\boldsymbol{\Sigma};\mathrm{skip}}
> = \mathrm{Cov}(\boldsymbol{y})(\mathrm{Cov}(\boldsymbol{y}) + \boldsymbol{\Sigma})^{-1},
> $$
>
> $$
> c_{\boldsymbol{\Sigma};\mathrm{out}}^{2}\boldsymbol{I}
> = \mathrm{Cov}(\boldsymbol{y})(\mathrm{Cov}(\boldsymbol{y}) + \boldsymbol{\Sigma})^{-1}\boldsymbol{\Sigma},
> $$
>
> $$
> \lambda_{\boldsymbol{\Sigma}}
> = \frac{1}{c_{\boldsymbol{\Sigma};\mathrm{out}}^{2}}.
> $$
>
> The matrix $C_{\Sigma;\text{noise}}$ is chosen empirically (as in EDM), and the scalar $\lambda_{\Sigma}$ is the corresponding loss weight.
>
>
> These expressions reduce to the EDM formulas in the isotropic case  $ \boldsymbol{\Sigma} = \sigma^{2}\boldsymbol{I} $
> when using $ \mathrm{Cov}(\boldsymbol{y}) = \sigma_{\text{data}}^{2} \boldsymbol{I} $. They also hold when $ \boldsymbol{\Sigma} $ is estimated from the data, provided the estimator is unbiased.
>
> In our main results, $ \boldsymbol{\Sigma} $ is anisotropic but diagonal, so we set $ \sigma_{\text{data}}^{2} = 1 $ after data normalization, following [1].
>
> **Reference**
>
> [1] Ilan Price, et al. Probabilistic weather forecasting with machine learning. *Nature*, **637**(8044):84–90, 2025.

---

### Author Response · Authors · 2025-12-03
**Official Comment for Area Chair**

Dear Area Chair and reviewers,

We thank the reviewers for their careful and constructive feedback. We are encouraged that the paper is seen as technically sound and relevant, with multiple reviewers noting that *“TEDM achieves strong accuracy on several long-horizon benchmarks”* (B111), *“provides a clear, modular extension of EDM to time-series”* (B111, rk7Q, r426), *“features well structured experiments, ablations, and complexity comparisons with clear qualitative plots”* (B111), and *“provides explicit quantitative evidence that the proposed framework can match or outperform diffusion-style baselines … while significantly reducing inference cost.”* (rk7Q). Reviewer r426 highlights *“exceptional computational efficiency”*, and reviewer 8F9D notes that TEDM *“demonstrates traits valuable for industrial/clinical adoption.”* Overall, the reviews regard TEDM as a promising contribution at the intersection of diffusion models and time-series forecasting.

TEDM (“Time Series Forecasting with Elucidated Diffusion Models”) is an autoregressive diffusion framework for multivariate long-horizon forecasting. It builds on EDM but aligns diffusion time with physical time and learns data-driven noise/scale schedules from time-series windows, yielding an ODE-based sampler with one Euler step per forecast step (O(H) complexity).

**Technical contributions & novelty**

* **Data-driven schedules:** Generalizes EDM to a multivariate setting with matrix-valued covariance Σₜ and scale sₜ, estimated empirically (cumulative/sliding windows) per physical time step instead of using hand-crafted scalar schedules.
* **Full-Σ treatment:** Extends EDM-style preconditioning to full Σₜ so the denoiser operates in a normalized space; in practice, a diagonal Σₜ approximation is used but is now explicitly derived and justified from the full-Σ formulation.
* **Time-aligned ODE:** Aligns diffusion and physical time, enabling deterministic probability-flow ODE sampling with a single network evaluation per step and autoregressive rollout over the forecast horizon.
* **Quantile sampler:** Introduces an ODE-based quantile estimation strategy for probabilistic forecasting that reuses the deterministic sampling.

**Empirical evidence**

* Evaluated on eight benchmarks (ETTh1/2, ETTm1/2, Exchange, Solar, Stock, Weather) at H = 96, with extended horizons up to 720 on ETTh2, ETTm2, Exchange; TEDM is strongest on several datasets compared to diffusion and non-diffusion SOTAs.
* Ablations show large gains over EDM/iDDPM+DDIM from empirical schedules; lightweight denoisers (even a linear layer) achieve low error with low latency and memory and moderate scaling in time/memory with horizon.

**Revisions after reviews**

During the discussion phase, we clarified all the reviewers concerns, especially around theory (diagonal covariance approximation, probabilistic performance) and experiments (additional baselines and datasets):

* **Theory expansion:** (B111, r426) We expanded the theory to full Σₜ (including preconditioning functions), clarified assumptions and the validity of the diagonal approximation, added additional experiments, and introduced an explicit limitations section (Appendix A, Sections 4.2–4.4).
* **Datasets and baselines:** (rk7Q, r426) We added Solar and Stock and included strong non-diffusion baselines (iTransformer, TimesNet, DLinear, PatchTST, Client) in the main comparison tables (Section 5, Table 3).
* **Efficiency, scalability, reproducibility, robustness:** (B111, rk7Q, r426, 8F9D) We reported runtime/memory benchmarks (updated Table 5, Figure 7), horizon-scaling plots (Appendix F), 4×-seed robustness (Appendix E), preconditioning functions and detailed hyperparameters for reproducibility (Table 6, Appendix B).
* **Probabilistic improvements:** (B111, rk7Q, r426, 8F9D) We analyzed weaknesses of the original SDE-based sampler and introduced the ODE-based quantile-flow scheme, which substantially improves CRPS and brings TEDM closer to NsDiff on key datasets (Appendix D).

We believe we have addressed all the reviewer concerns with targeted theoretical and empirical updates. The additional results in the revised manuscript, together with improved presentation, figures, tables, and analysis, are consistent with and reinforce our original findings. They support TEDM as a principled, memory-efficient diffusion framework that helps to elucidate the design space of diffusion models for time-series forecasting.

Thank you again for your time and service to the community.

Best regards,

Authors

---

### Meta-Review · Area_Chair_ydpN · 2026-01-08

**Summary:**

Reviewers generally found the paper to be technically strong and well motivated, with a principled extension of the EDM framework to time-series forecasting and a novel alignment of diffusion and physical time that significantly reduces inference complexity. The empirical evaluation demonstrated strong forecasting accuracy across multiple long-horizon benchmarks and included detailed efficiency analyses.

The main concerns focused on theoretical assumptions (in particular the diagonal covariance approximation and its implications), probabilistic forecasting quality in comparison to strong diffusion baselines, clarity and completeness of methodological details affecting reproducibility, and comparative positioning, including the relationship to closely related diffusion-based approaches and the absence of some non-diffusion baselines in the initial submission

**Reviewer Concerns:**

The rebuttal and revision address many of the reviewer concerns by clarifying the theoretical formulation, including the role and implications of the diagonal covariance approximation, expanding methodological details to improve clarity and reproducibility, and strengthening the empirical evaluation. In particular, the authors added non-diffusion baselines, additional datasets and horizon analyses, multi-seed results, and explicit runtime and memory comparisons. They also improved the treatment of probabilistic forecasting by introducing an ODE-based quantile estimation strategy and clarifying the relationship between TEDM and closely related diffusion-based methods.

Some concerns remain partially addressed or naturally limited by scope. While the assumptions underlying the diagonal covariance approximation are now better motivated and empirically explored, their behavior in more challenging high-dimensional settings remains an open question. Probabilistic forecasting quality has been improved, but still relies on additional modeling choices whose full implications could be further explored. Overall, these remaining issues appear to reflect directions for future work rather than gaps in the core validation of the proposed approach.

**Reviewer Scores:**

Even assuming a full discussion, it is difficult to assess how individual reviewers would have adjusted their scores. The two reviewers with scores of 4 may have increased their scores slightly, as their main theoretical, empirical, and clarity-related concerns were directly addressed in the rebuttal. The two reviewers with scores of 6 are less likely to substantially change their evaluations. Overall, a hypothetical reviewer–author discussion would have likely narrowed the initial slight disagreement and strengthened the consensus.

---

### Decision · Program_Chairs · 2026-01-26

Accept (Poster)